1    **Dry and warm conditions in Australia exacerbated by aerosol reduction in China**

6    Jiyuan Gao[1], Yang Yang[1*], Hailong Wang[2], Pinya Wang[1], Hong Liao[1]

[1]Joint International Research Laboratory of Climate and Environment Change (ILCEC), Jiangsu
Key Laboratory of Atmospheric Environment Monitoring and Pollution Control, Jiangsu
Collaborative Innovation Center of Atmospheric Environment and Equipment Technology,
School of Environmental Science and Engineering, Nanjing University of Information Science
and Technology, Nanjing, Jiangsu, China
[2]Atmospheric, Climate, and Earth Sciences Division, Pacific Northwest National Laboratory,
Richland, Washington, USA

26    *Correspondence to yang.yang@nuist.edu.cn

**Abstract**

A substantial decline in anthropogenic aerosols in China has been observed since the initiation of clean air actions in 2013. This study reveals a linkage between aerosol reductions in China and Australia's drier and warmer conditions. Aerosol decline in China triggered alterations in temperature and pressure gradients between the two hemispheres, leading to intensified outflow from Asia towards the South Indian Ocean, strengthening the Southern Indian Subtropical High and its related Southern Trade Winds. Consequently, this atmospheric pattern resulted in a moisture divergence over Australia. The reduction in surface moisture further resulted in more surface energy being converted into sensible heat instead of evaporating as latent heat, warming the near-surface air. The intensified dry and warm climate conditions further caused the increase in wildfire risks during fire seasons in Australia. Our study illuminates the impact of distant aerosols on precipitation and temperature variations in Australia, offering valuable insights for drought and wildfire risk mitigation in Australia.

**1 Introduction**
Australia encompasses various climate zones, ranging from the tropical climate in the north
to arid conditions in the interior and temperate climates in the south (Head et al., 2014). The
continent is predominantly dry, receiving an average annual rainfall of less than 600 mm and less
than 300 mm over half of the land. Evident long-term trends can be observed in Australia's
historical rainfall records. These trends reveal a notable shift towards drier conditions across
southern Australia (Dey et al., 2019a; Nicholls, 2006; Rauniyar and Power, 2020; Wasko et al.,
2021), and an increase in rainfall before 2010 (Dey et al., 2019a, b; Evans et al., 2014; Nicholls,
2006; Rotstayn et al., 2007; Wasko et al., 2021) followed by a slight decreasing trend of rainfall
after 2010 (CSIRO and BOM, 2022) in the northern Australia.
Precipitation in Australia is influenced by a variety of atmospheric circulation systems,
including East Coast Lows (ECLs), the Australian-Indonesian Monsoon, tropical cyclones (TCs),
fronts, and different modes of large-scale climate variabilities, such as the El Niño-Southern
Oscillation (ENSO), Indian Ocean Dipole (IOD), Interdecadal Pacific Oscillation (IPO),
Subtropical ridge (STR), Southern Annular Mode (SAM), and Madden Julian Oscillation (MJO)
(Dey et al., 2019a; Risbey et al., 2009). The linkages between Australia's rainfall characteristics
and these drivers could change in response to the internal natural variabilities and external
anthropogenic forcings. Rauniyar and Power (2020) reported that the drier conditions across the
southern Australia could be attributed to a combination of both decadal-scale natural variability
and changes in large-scale atmospheric circulation patterns, which was linked to the escalating
emissions of greenhouse gases (GHGs), while Rotstayn et al. (2007) found that the increased levels
of rainfall in northern Australia before 2010 was linked to the increases in aerosols in Asia.
Human activities have led to a rise of global surface air temperature by approximately 1.29 °C
(0.99 to 1.65 °C) from 1750 to 2019, mainly due to an enhanced greenhouse effect from increasing
GHG emissions (IPCC, 2021). In addition to GHGs, human activities also emit a variety of
aerosols and their gaseous precursors into the atmosphere. Since industrialization, there has been
a significant rise in the levels of aerosols and precursors (Hoesly et al., 2018). Atmospheric
aerosols are the second-largest anthropogenic climate forcer, exerting an overall cooling effect that
partially masks the warming induced by GHGs. (IPCC, 2013, 2021). However, as anthropogenic
aerosols declined during the past decades in many countries of the world related to the clean air
actions, the associated "unmask" effect is likely to exacerbate GHG-induced warming (Kloster et
al., 2010). For example, in the 1980s, clean air actions were implemented in North America and
Europe, leading to a decrease in the emissions of aerosols and their precursors (Hoesly et al., 2018).
Reductions in aerosol emissions in the U.S. have led to changes in aerosol direct radiative forcing
(DRF) by 0.8 W m$^{-2}$ and indirect radiative forcing (IRF) by 1.0 W m$^{-2}$ over the eastern U.S. during
1980–2010 (Leibensperger et al., 2012). Similarly, aerosol decreases over Europe between the
1980s and 1990s have caused a change in regional DRF by 1.26 W m$^{-2}$ (Pozzoli et al., 2011). In
China, the emissions of aerosols and their precursors have been reduced since 2013 due to the
implementation of Air Pollution Prevention and Control Action Plan. Dang and Liao (2019)
reported a 1.18 W m$^{-2}$ change in DRF between 2012 and 2017 due to decreased aerosol levels over
eastern China. Gao et al. (2022) estimated a warming of 0.20 °C in China, 0.15 °C in North
America, and 0.14 °C in Europe, attributed to the decreases in aerosols during 2013–2019.
Monsoonal rainfall serves as a vital resource for agriculture, industry, and ecosystems across
the monsoon-affected regions, affecting approximately two-thirds of the world's population (B.
Wang et al., 2021). Apart from GHGs-induced warming (Cook and Seager, 2013) and nature
variabilities (e.g., ENSO) (Oh and Ha, 2015), aerosol also modulates monsoon system mainly
through changing land-sea temperature and pressure gradient. The impact of aerosols on Asian
monsoon has been widely investigated. Based on climate model simulations, Liu et al. (2023)
found that aerosol reductions in East Asia during 2013–2017 resulted in an approximately 5%
increase in the strength of the East Asian summer monsoon (EASM). The EASM is also reported
to enhance due to future aerosol reductions from 2000 to 2100 (Wang et al., 2016). Dong et al.
(2019) explored the effects of increased aerosol in Asia from 1970s to 2000s on atmospheric
circulation and rainfall patterns and found anomalous moisture convergence and increased
precipitation over the Maritime continent. Non-Asian aerosols also have an effect on South Asian
monsoon rainfall through changes in the interhemispheric temperature gradient and meridional
shifts of the Intertropical Convergence Zone (Bollasina et al., 2011, 2014; Cowan and Cai, 2011;
Undorf et al., 2018). Australia, especially in northern Australia, is largely affected by the Australian
monsoon, which is characterized by winds that blow from the southeast during cold season and
from the northwest during the warm season (Gallego et al., 2017; Heidemann et al., 2023).
Australia has a relatively low level of anthropogenic aerosols, which suggests that the impact of
domestic anthropogenic aerosols on Australian monsoon should not be significant. However, the
impact of remote aerosols on Australian monsoon have been investigated in previous studies, and
they reported that the increases in Asian aerosols could enhance rainfall in Australia through
increasing monsoonal winds towards Australia (Rotstayn et al., 2007; Fahrenbach et al., 2023).
Wildfires, which are uncontrolled fires spreading rapidly across natural landscapes, are
significantly influenced by meteorological conditions (He et al., 2019; Jones et al., 2022). In
Australia, these fires, also known as bushfires, present a major environmental and social threat
(Johnston et al., 2021; Ward et al., 2020). According to Dowdy (2020), the most favorable seasons
for wildfires in most regions of Australia are austral spring and summer. Key meteorological
factors such as extended periods of drought, elevated temperatures, low humidity, and strong winds
are crucial in determining the occurrence and intensity of wildfires (Zacharakis and Tsihrintzis,
2023). Under the recent historical climate change, Australia has witnessed a rise in extreme fire
weather conditions and an extended fire season (CSIRO and BOM, 2022).
Since the implementation of clean air policies in 2013, there has been a noticeable decrease
in aerosol levels in China (Zhang et al., 2019; Zheng et al., 2018). Numerous studies have shown
the local and global climate effects of the aerosol reductions in China (Dang and Liao, 2019; Gao
et al., 2022, 2023; Liu et al., 2023; Zheng et al., 2020). The Australian monsoonal wind patterns,
reported to be influenced by Asian aerosol emissions in the past decades (Rotstayn et al., 2007;
Fahrenbach et al., 2023), may have also influenced by the radiative effects due to the aerosol
decline in China. The objective of this study is to explore how aerosol emission reductions in China
affect the climate and wildfire risk in Australia and to investigate the underlying mechanisms
driving these impacts.
**2 Methods**
**2.1 Observational and Reanalysis Data**
Ground-based observational data of near-surface $PM_{2.5}$ concentrations in China 2013–2019
are acquired from the China National Environmental Monitoring Centre (CNEMC), which offers
daily records of near-surface air pollutant concentrations for nearly 1800 sites. Aerosol Optical
Depth (AOD) data are obtained from the Moderate Resolution Imaging Spectroradiometer
(MODIS) Deep Blue retrieval (Hsu et al., 2013). These observational data are used for evaluating
the performance of model simulated aerosols.
ERA5, the fifth generation of the European Centre for Medium-range Weather Forecasts
(ECMWF) atmospheric reanalysis, is a comprehensive dataset that provides a detailed and globally
consistent view of the earth's atmospheric conditions over the past several decades (Hersbach et
al., 2020). In this study, ERA5 data are employed to evaluate the climate condition in Australia
and analyze its linkage with aerosol reductions in China during the 2013–2019 by using the 2-
meter temperature, total precipitation, relative humidity, cloud cover, wind fields, vertical velocity,
surface latent and sensible heat flux, and surface and top of the atmosphere (TOA) solar and
longwave radiative flux under both clear and all sky conditions.
The Global Precipitation Measurement (GPM) mission is a joint initiative by National
Aeronautics and Space Administration (NASA) and the Japan Aerospace Exploration Agency
(JAXA) aimed at providing accurate and frequent measurements of global precipitation
(Skofronick-Jackson et al., 2017). GPM includes a core satellite equipped with advanced radar and
microwave sensors, enabling the observation of rain and snowfall in real-time. The data are crucial
for understanding weather patterns, climate dynamics, and hydrological processes. The Global
Precipitation Climatology Project (GPCP) is an international initiative under the World Climate
Research Programme (WCRP) that integrates satellite and gauge-based observations to provide
long-term, global precipitation estimates (Adler et al., 2018). GPCP offers monthly and daily
precipitation datasets with global coverage, combining data from rain gauges, satellite infrared
(IR), and microwave sensors to enhance accuracy. Precipitation rate data from GPM and GPCP
are also used in the study. GPM provides higher temporal and spatial resolution data compared to
Global Precipitation Climatology Project (GPCP), making it more suitable for studies focused on
short-term precipitation variability and regional climate dynamics.
**2.2 Model Description and Experimental Design**
In this research, we conduct simulations to explore the impact of aerosols on climate using
the Community Earth System Model version 1 (CESM1). CESM1 simulates the major aerosols
including sulfate, black carbon (BC), primary organic matter (POM), secondary organic aerosol
(SOA), dust and sea salt in a four-mode Modal Aerosol Module (MAM4), as described in Liu et
al. (2016). CESM1 simulations are carried out with 30 vertical layers and a horizontal resolution
of 2.5° longitude by 1.9° latitude. In addition to the default model physics, several supplementary
features are incorporated into the model in this study to improve the model's performance in
simulating aerosol wet scavenging and convective transport (Wang et al., 2013).

163        The global anthropogenic emissions of aerosols and their precursors are obtained from the

Community Emissions Data System (CEDS) v_2021_04_21. In contrast to the prior CEDS
v_2016_07_26, which exhibits significant regional emission biases (Z. Wang et al., 2021), the
newer CEDS version of anthropogenic emissions of aerosols and precursors considers the
substantial reductions in emissions in China, related to the recent clean air actions since 2013 (Fig.
S1). Specifically, anthropogenic sulfur dioxide ($SO_2$), BC, and organic carbon (OC) emissions
decreased by −12.48, −0.30, and −0.21 Tg yr$^{-1}$, respectively, over China between 2013 and 2019.
Biogenic emissions are from the Model of Emissions of Gases and Aerosols from Nature version
2.1 (MEGAN v2.1) (Guenther et al., 2012), while the emissions from open biomass burning are
derived from the CMIP6 (Coupled Model Intercomparison Project Phase 6) (Van Marle et al.,

173    2017).

174        A series of model experiments are conducted using CESM1 with a fully-coupled model

configuration, as detailed in Table S1. In the baseline scenario (referred to as BASE),
anthropogenic emissions of aerosols and precursors are fixed at year 2013 worldwide. In CHN,
anthropogenic emissions of aerosols and precursors over China are fixed at year 2019, while
emissions in all other regions are remained at year 2013. In NAEU, the simulation is performed
with anthropogenic emissions of aerosols and precursors over North America and Europe set at
year 2019, while emissions in other regions remained at year 2013. In OTH, anthropogenic
emissions of aerosols and precursors in other regions except for China are set at year 2019, while
emissions in China are kept at year 2013. In SASEA, anthropogenic emissions of aerosols and
precursors over South Asia and Southeast Asia are fixed at year 2019, while emissions in other
regions remained at year 2013. Biogenic and biomass burning emissions worldwide in all
experiments are fixed at year 2013. To reduce model biases related to internal variability, three
ensemble members are conducted by perturbing the initial atmospheric temperature conditions.
All simulations are run for 150 years, with the last 100 years for detailed analysis. To additionally
examine the fast climate responses to aerosol reductions in China, supplementary atmosphere-only
experiments (BASE_FAST and CN_FAST) were conducted using the CESM1 atmospheric
component (CAM5) with prescribed sea surface temperatures and sea ice concentrations. These
simulations were run for 30 years, and the outputs from the last 15 years were analyzed.

**2.3 Model Evaluation**

To validate whether CESM1 can reproduce the aerosol reductions in China during the 2010s,
changes in simulated near-surface $PM_{2.5}$ concentrations (sum of sulfate, BC, POM, SOA, dust×0.1,
and sea-salt×0.25 following Turnock et al. (2020)) in China during 2013–2019 are compared with
the observations. Figure S2 shows spatial distributions of observed and modeled annual mean near-
surface $PM_{2.5}$ concentration changes over China (2017–2019 minus 2013–2015), which exhibited
a statistically significant correlation coefficient of 0.52 between simulations and observations.
However, the model underestimates the $PM_{2.5}$ concentration changes by 76% in China. The
considerable underestimation has been reported in many previous studies, resulting from coarse
model resolution,  uncertainties in emissions of aerosols and precursor gases, strong aerosol wet
removal, and the model's deficiency in simulating nitrate and ammonium aerosols (Fan et al., 2018,
2022; Gao et al., 2022, 2023; Zeng et al., 2021). The model has a good capability in replicating
the spatial distribution of AOD changes in China during 2013–2019 (Fig. S3), as evidenced by a
high correlation coefficient of 0.83, but the model also exhibits an underestimation in the AOD
reductions by 69%. However, the apparent underestimation of absolute changes may largely stem
from the inherently low background aerosol concentrations in CESM1. When considering the
relative changes rather than absolute values, the bias appears much less pronounced (Fig. S4 and
Fig. S5).
Climate variables, including precipitation rate, surface air temperature, relative humidity,
total cloud cover, surface solar radiation, 10m wind speed, and surface and TOA net total radiative
fluxes under both clear and all sky conditions over Australia simulated by CESM1 model are also
compared with those from ERA5 reanalysis (Fig. S6–S8). The model demonstrates a good
performance in simulating Australian climate, with normalized mean bias (NMB) values
consistently below or near 40% for surface air temperature, relative humidity, total cloud cover,
surface downward solar radiation, and 10m wind speed, and surface and TOA net total radiative
fluxes, but it tends to overestimate annual precipitation by about 90%, especially over coastal
regions likely related to the coarse model resolution. The model accurately reproduces spatial
patterns of all climate variables, closely aligning with observations, as indicated by correlation
coefficients ranging from 0.7 to 1.0.
**2.4 Wildfire Risk Indices**
In this study, several climatological indices are used to indicate wildfire risk during fire
seasons (austral spring and summer, from September to February of the next year) in Australia
(Ren et al., 2022; Irmak et al., 2003; Seager et al., 2015; Sharples et al., 2009).
(i) Reference Potential Evapotranspiration ($ET_0$):
$ET_0$ is a climatological index used to estimate the amount of water that could potentially
evaporate and be transpired from the Earth's surface under specific meteorological conditions. The
calculation of $ET_0$ takes into account factors such as temperature (T, unit: °C), and surface
downward solar radiation ($R_s$, unit: W m$^{-2}$) to estimate the maximum amount of water loss due to
evaporation and transpiration. $ET_0$ is important in wildfire studies because it helps to gauge the
environmental moisture conditions and the potential for drought, which can be a significant factor
in wildfire risks assessment. $ET_0$ is given by the following expression (Irmak et al., 2003):
$$ET_0 = -0.611 + 0.149R_s + 0.079\text{T}$$

(ii) Vapor Pressure Deficit (VPD):
Vapor Pressure Deficit (VPD) is a meteorological parameter that measures the difference
between the amount of moisture in the air and the maximum amount of moisture the air can hold
at a given temperature (T, unit: °C) and moisture (relative humidity, RH, unit: %). High VPD
values indicate that the air is dry. VPD is important in the context of wildfires because it reflects
the drying potential of the atmosphere. When VPD is high, it can lead to rapid moisture loss from
vegetation, making it more susceptible to ignition and increasing the risk of wildfires. VPD is
given by (Seager et al., 2015):
$$VPD = \frac{100 - RH}{100} \times 610.7 \times 10^{\frac{7.5T}{237.3+T}}$$

(iii) McArthur Forest Fire Danger Index (FFDI):
The McArthur Forest Fire Danger Index (FFDI) is a widely used index in Australia to assess
the potential for bushfires and forest fires. It takes into account various meteorological factors,
including T (unit: °C), RH (unit: %), wind speed (U, unit: m s$^{-1}$), and drought factor (DF, unitless).
We set DF as 10 here following Sharples et al. (2009), as it would not significantly affect the
methods of comparison used later in their study. While we acknowledge that this assumption is
idealized, we find it applicable in our case. To verify this, we calculated gridded DFs for Australia
(Figure S9), which show that the DFs are close to 10 and exhibit nearly homogeneous spatial
distributions. As such, setting DF = 10 for Australia is reasonable for our analysis. In addition,
FFDI calculated using DF = 10 and FFDI using gridded DFs are compared (Figure S10). The
patterns and regional averages of both datasets are very similar, further supporting the use of DF
= 10 in our study. The FFDI provides a numerical rating that indicates the level of fire danger,
with higher values corresponding to greater fire risks. This index is particularly valuable for
assessing the immediate risk of wildfires and is commonly used in fire management and prediction.
FFDI is defined as (Sharples et al., 2009):
$$FFDI = 2e^{-0.45+0.987lnDF+0.0338T-0.0345RH+0.0234U}$$
**3 Results**
**3.1 Intensified Dry and Warm Conditions in Australia by aerosol changes**
Figure 1 shows simulated responses in annual precipitation rate, surface air temperature and
relative humidity in Australia to changes in anthropogenic emissions of aerosols and precursors.
In response to aerosol reductions in China, Australia experiences significant decreases in
precipitation and relative humidity, while the temperature has an increase from 2013 to 2019. On
regional average, annual precipitation, surface air temperature and relative humidity change by –
0.10 mm day$^{-1}$, 0.06 °C, and –1.20%, respectively, in Australia caused by the aerosol reduction in
China during this time period, contributing to the dry and warm climate in Australia. Notably,
Northern Australia experiences the most significant reduction in convective precipitation, whereas
Southern Australia has the greatest decline in large-scale precipitation related to the aerosol
reduction in China, as simulated by the CESM1 model (Fig. S11). The direction of seasonal
responses in precipitation rate, surface air temperature and relative humidity are the same as the
annual averages, with the largest changes occurring in austral spring (Fig. 2). Regarding the strong
seasonal component observed in the responses, there are two main contributing factors: (1) the
background seasonality of the Southern Hemisphere trade winds, and (2) the seasonality of China's
emission changes themselves. However, determining which of these factors dominates requires
further detailed quantitative analysis.
The intensified dry and warm conditions in Australia can also be seen in the observations,
as indicated by ERA5 reanalysis data (Fig. 3).  Since 2010, precipitation and relative humidity
have significantly decreased in Australia, especially in Northern and Eastern Australia, at a rate of
0.086 mm day$^{-1}$ yr$^{-1}$ and 1.07% yr$^{-1}$, respectively, while surface air temperature has increased at a
rate of 0.17 °C yr$^{-1}$. In addition, Time series of precipitation have also been extended to cover
2001–2019 (Fig. S12). To account for the potential influence of internal variability, particularly
ENSO, which is a major driver of Australia's precipitation variability (Fig. S13), the time series
are also given after removing the influence of Niño 3.4 index. The adjusted results exhibit a similar
drying trend since the 2010s (Fig. S12). The GPM and GPCP data also exhibit a similar decreasing
trend in precipitation over Australia (Fig. S14). The reduction in anthropogenic $SO_2$ emissions in
China shows strong correlations with the decrease in precipitation and the increase in temperature
in Australia during 2010–2019 (Fig. S15). However, when extending the time frame to the period
before emissions reductions in China (1940–2019), the increase in temperature becomes less
pronounced, with a slight rise in precipitation and relative humidity, likely attributed to greenhouse
gas warming, which can serve as indication that the decrease in precipitation and increase in
temperature in Australia from 2010 to 2019 are not primarily caused by GHGs (Fig. S16). To place
the recent drying trend in a longer-term context, all 10-year precipitation trends in ERA5 data from
1940 onwards were calculated. The 10-year trends (unit: mm day$^{-1}$ year$^{-1}$) were found to be +0.016
(1940–1949), –0.012 (1950–1959), +0.001 (1960–1969), +0.031 (1970–1979), –0.002 (1980–
1989), +0.030 (1990–1999), –0.032 (2000–2009), and –0.086 (2010–2019). These results indicate
that the drying trend during 2010–2019 was unusually large compared to other decades since 1940,
suggesting that despite the influence of internal variability, a pronounced reduction in precipitation
over Australia has been observed in recent years. The rainfall decrease is consistent with changes
in clouds. Spatial distributions of simulated changes in vertically-integrated cloud cover are shown
in Figure S17, indicating a reduction in clouds of all levels, including high, mid-level, and low
clouds. In addition, the spatial distributions of these changes closely resemble the patterns of
responses in precipitation and relative humidity.
Aerosol emissions have changed globally during 2013–2019, not only in China but also in
regions such as South Asia, Southeast Asia, North America, Europe, and Australia, affecting
climate both locally and remotely. Figure 1 shows the changes in climate variables in Australia
resulting from aerosol emission changes in these regions. Emission changes in other regions except
China yield an overall decrease in precipitation by 0.06 mm day$^{-1}$, an increase in temperature by
0.06 °C, and a decrease in relative humidity by 0.67%. Among them, aerosol emission changes in
South Asia and Southeast Asia yield the largest responses, with a decrease in precipitation by 0.08
mm day$^{-1}$, an increase in temperature by 0.06 °C, and a decrease in relative humidity by 0.87%.
Emission changes in North America and Europe contribute to a decrease in precipitation by 0.04
mm day$^{-1}$, an increase in temperature by 0.04 °C, and a decrease in relative humidity by 0.41%,
although these responses are mostly insignificant (Fig. 1).
**3.2 Mechanisms of Dry and Warm conditions in Australia Amplified by Aerosol Reductions**
**in China**
The rising levels of Asian aerosols could influence the meridional temperature and pressure
gradients across the Indian Ocean and therefore affect monsoonal winds and rainfall in Australia
since the middle of the 20[th] century, as reported in several previous studies (Fahrenbach et al.,
2023; Rotstayn et al., 2007). Since 2013, aerosol levels in China have substantially decreased due
to clean air actions initiated by the Chinese government (Zhang et al., 2019). At the same time,
precipitation in Australia exhibited a declining trend, which could be partly attributed to the
decrease in China's anthropogenic aerosol forcing as quantified through CESM1 simulations.
Asian monsoon region is closely connected with the meridional Hadley circulation and zonal
Walker circulation through monsoon outflow to the South India ocean subtropical high (SISH) and
North Pacific subtropical high (NPSH) (Beck et al., 2018). Figure 4a illustrates the climatological
mean wind fields at 850 hPa, indicating the persistent existence of SISH in the Indian Ocean and
NPSH in the North Pacific. With reductions in aerosols in China, the sea surface temperature (SST)
increases in the North Pacific but decreases in the Indian Ocean (Fig. 4b), which is concurrent with
the northward shift of the Intertropical Convergence Zone (ITCZ) (Basha et al., 2015). Over Asia,
this migration of ITCZ is accompanied by the northward movement of the upper-tropospheric
subtropical zonal westerly jet (Chiang et al., 2015; Schiemann et al., 2009), which moves to the
north of the Tibetan Plateau. It then enhances the circulation pattern of the Tibetan high, redirects
the outflow from the Asian monsoon to the southern Indian Ocean subtropical high (Fig. 5c),
strengthens the SISH, and leads to the enhancement of the Southern Trade Winds (Fig. 5d). On
the other hand, the increase in SST in the North Pacific induces ascending motion around the 130°–
150°E and the subsequent descending motion around the 90°–110°E (Fig. 5b), with anomalous
westerly winds near the surface, leading to a weakening of NPSH along with a decrease in the
Northern Trade Winds (Fig. 5d). Note that, the descending motion partly compensates the
ascending motion related to the meridional circulation between 10°–30°N (Fig. 5c). The
mechanism of China's aerosol reductions on the large-scale 3D circulation in the Asia-Pacific
region is shown in Figure 6. The enhancement of the Southern Trade Winds further causes
moisture advection away from Australia, accompanied by moisture divergence in Australia,
especially over the northern Australia (Fig. S18). This moisture divergence in Australia then
intensifies the dryness in Australia (Fig. 1).

346       Figures 7 illustrate the changes in relevant radiative fluxes in Australia resulting from
aerosol changes in China. Under the clear sky condition, both surface and top of the atmosphere
(TOA) radiative flux decrease (Fig. 7c&d). It is due to increased sea salt and dust aerosols (Fig.
S19b–d) due to the stronger Southern Trade Winds (Figs. S19a and 5d) and dryer conditions (Fig.
1). The decrease in cloud cover (Fig. S17) leads to an overall increase in both surface and TOA
radiative flux (Fig. 7e&f). The overall changes in radiative fluxes are offset by these two factors
(Fig. 7a&b) and insufficient to explain the significant increase in surface air temperature in
Australia (Fig. 1).

354       Decreases in precipitation lead to a decrease in surface specific humidity (Fig. S20a), which
declines more than that at 850 hPa (Fig. S20b). This results in excess energy being converted into
sensible heat rather than latent heat through evaporation (Chiang et al., 2018; Fischer et al., 2007;
Seneviratne et al., 2006; Su et al., 2014), which is indicated by a decrease in surface upward latent
heat flux and an increase in surface upward sensible heat flux in Australia due to aerosol changes
in China in Figure 8. The increased surface upward sensible heat flux heats the near-surface air
and contributes to the warm conditions in Australia.

### 3.3 Increases in Wildfire Risk in Australia

Wildfires represent a biosphere-atmosphere phenomenon, arising from the intricate interplay of weather, climate, fuels, and human activities (Moritz et al., 2014). Notably, wildfires are ranked among the most significant natural disasters in Australia, causing extensive damage (Shi et al., 2021). Collins et al. (2022) reported that warmer and drier conditions increased the potential for large and severe wildfire in Australia. Given that changes in aerosols in China have led to a warmer and drier climate condition in Australia in recent years, the change in this climate state could also impact on the occurrence of wildfires. Three wildfire risk indices ($ET_0$, VPD, and FFDI) are selected to assess the risks of wildfires occurrence. Detailed information about the three wildfire risk indices can be found in the Methods section.

All three indices exhibit increases (+0.37 mm mon$^{-1}$ for $ET_0$, +0.56 hPa for VPD, and +0.25 for FFDI) during fire seasons (September to February) in Australia due to changes in aerosols in China during 2013–2019 (Fig. 9), although they do not show the same spatial distribution possibly due to different considerations regarding climate variables in the indices.

### 4 Conclusions and Discussions

This study reveals that the substantial aerosol reduction in China can lead to drier and warmer conditions in Australia. Aerosol reductions in China induce changes in temperature and pressure gradients, which lead to an increased outflow from Asia towards the South Indian Ocean, strengthening the SISH and the associated Southern Trade Winds. Consequently, this atmospheric pattern results in moisture divergence over Australia, causing a decrease in humidity and precipitation. The reduction in surface moisture leads to more surface energy being converted into sensible heat, rather than evaporating as latent heat, thereby heating the near-surface air. This perspective sheds light on the influence of distant aerosols on climate in Australia.

The CESM1 simulations depict warmer and drier conditions in Australia related to China's aerosol reductions than otherwise, a pattern also evident in observations represented by ERA5. It should be noted that the short-term precipitation trends examined in this study may be influenced by internal climate variability, especially given the relatively short period since 2013. Therefore, we extended the analysis back to around 2010, when emission reductions in China had already

begun. We also performed additional analysis examining different time periods, including the
2013–2019 period (Fig. S21) and a longer period (2010–2023; Fig. S22). Both patterns
consistently show a decreasing trend in Australia's precipitation, supporting the robustness of the
recent drying trend in Australia. However, we acknowledge that these observed short-term trends
could still be affected by internal climate variability, and the potential contribution from internal
variability should not be ignored.
Although aerosol reductions may contribute to worsened climate conditions in Australia, air
quality and human health improvements owing to aerosol reductions cannot be ignored (Giani et
al., 2020; Zheng et al., 2017). In addition, the drier and warmer conditions in Australia  linked to
aerosol reductions should be considered resulting from the rise in long-lived GHGs, while the
aerosol reductions unmask the effect rooted in the GHGs (Wang et al., 2023). In addition, other
factors such as internal climate variability (Heidemann et al., 2023) appear to have contributed
more to the changes in Australia's climate conditions.
Rotstayn et al. (2007) and Fahrenbach et al. (2023) investigated the impact of increasing
anthropogenic aerosols over Asia during the late 20th century and early 21st century and indicated
that enhanced aerosol emissions strengthened the meridional temperature and pressure gradients,
intensifying the monsoonal circulation and leading to increased precipitation over Australia. In
contrast, our study focuses on the reductions in anthropogenic aerosols in China, showing that the
aerosol reductions reversed circulation changes induced by the previous aerosol increases and led
to drying rather than increased rainfall over Australia. Our study also explores the subsequent
impact of aerosol reductions in China on Australian temperature. We find that reduced moisture
availability over Australia led to a shift in surface energy partitioning, with more energy being
converted into sensible heat rather than latent heat, resulting in near-surface warming. In summary,
while previous studies have demonstrated the role of increasing Asian aerosols in enhancing
Australia's precipitation, our work complements these findings by showing that decreasing
aerosols, particularly from China, can drive the opposite effect, leading to both drier and warmer
conditions in Australia.
Liu et al. (2018) used multi-model simulations to investigate the fast and slow precipitation
responses to regional aerosol forcing from Asia and Europe under the PDRMIP framework. They
showed that Asian sulfate aerosols are strong drivers of global temperature and precipitation
changes, with fast precipitation responses scaling with atmospheric absorption and slow responses
scaling with surface temperature. However, their analysis focused on aerosol forcing from the
entire Asian region, rather than emission reductions from China alone, and the precipitation
responses over Australia were not significant in their results. Hwang et al. (2024) explored the
combined fast and slow climate responses to aerosol emission changes at the global scale. They
highlighted that fast adjustments of atmospheric circulation and joint feedback between low clouds,
wind, evaporation, and sea surface temperature, followed by slower oceanic responses, can sustain
La Niña-like cooling patterns in the equatorial Pacific for decades. However, neither study
specifically examined the climate impacts of China's recent emission reductions.
We acknowledge that fast and slow climate responses can have different mechanisms and
regional impacts, and there can be differences between the slow and fast responses. Both fast and
slow climate responses contribute to the simulated precipitation changes in Australia resulting
from China's emission reductions. Fast responses are primarily driven by rapid atmospheric
adjustments to aerosol-induced radiative forcing without requiring full ocean adjustment, while
slow responses involve gradual changes mediated by ocean circulation and sea surface temperature
adjustments over longer timescales. In our additional fast response experiments using the CAM5
atmospheric component, we found a decreasing pattern in Australia's precipitation (Fig. S23),
which is consistent with the long-term equilibrium simulation results. Although the equilibrium
simulations represent the fully adjusted climate system response and are not appropriate to be
directly compared to short-term observational changes, the fast response experiments suggest that
fast response has contributed 86% to the fully climate response in the changing precipitation in
Australia. It demonstrates that the immediate atmospheric response to Chinese aerosol reductions
alone can produce a drying effect similar to that found in long-period equilibrium experiments,
providing confidence in the robustness of the results.
One limitation of this study is the use of equilibrium experiments. Although the equilibrium
method is useful for isolating aerosol-induced climate responses, it does not fully capture the
transient climatic evolution. Attributing observed climate changes directly to modeled aerosol-
induced responses involves inherent uncertainties. Given the uncertainties, here we did not attempt
to quantitatively attribute the observed climate change to aerosols in this study, but showed the
mechanisms that how aerosol decline influence remote climate. Future work could also consider
transient simulations, which can better represent the role of aerosols in the climatic evolution of
the real world.
There are some other potential uncertainties in this study. Firstly, the low bias in simulated
aerosol concentrations in CESM1 and CESM1's larger sensitivity to aerosol forcing could
introduce some uncertainties. Secondly, our findings are derived from simulations conducted with
a single fully-coupled climate model, and it is vital for future research to employ multi-model
ensemble simulations to reduce the possibility of model-dependent specific results. While the
CMIP6 and PDRMIP (Precipitation Driver Response Model Intercomparison Project) can assist
in minimizing such model dependencies, they also present certain drawbacks. Anthropogenic
emissions input in CMIP6 inadequately accounts for aerosol reductions resulting from clean air
actions in China since 2013 (Z. Wang et al., 2021). Additionally, CMIP6 considers aerosol changes
globally, making it challenging to isolate effects specifically induced by changes in aerosols in
China. The experimental design of PDRMIP, which scales the concentrations of sulfate and BC in
Asia to ten times of their present-day levels, is generally idealistic and may not accurately and
proportionally represent aerosol changes observed in the real world. Finally, in addition to aerosols,
GHGs and nature variabilities also contribute to regional and global climate change. The extent to
which GHGs and nature variabilities could contribute to weather condition and climate changes in
Australia remains unknown, which warrants further investigation.
Nonetheless, our study examined how China's aerosol reductions could affect Australia's
climate condition changes. Substantial emission reductions will continue in China following the
carbon neutral pathway (Yang et al., 2023), while future changes in emissions in South Asia remain
uncertain (Samset et al., 2019). Apart from natural climate variabilities that affect the Australian
monsoon, further investigation of changes in monsoon precipitation in Australia should also
consider the effects of remote aerosol changes simultaneously, which is crucial to effective drought
and wildfire management and mitigation in Australia.

**Acknowledgments**

This study was supported by the National Natural Science Foundation of China (grant no.
42475032), the Jiangsu Science Fund for Carbon Neutrality (grant no. BK20220031), Jiangsu
Innovation and Entrepreneurship Team (grant no. JSSCTD202346), and Postgraduate Research
Practice Innovation Program of Jiangsu Province (Grand KYCX25_1669). The Pacific Northwest
National Laboratory (PNNL) is operated for DOE by the Battelle Memorial Institute under contract
DE-AC05-76RLO1830.
**Data and Code Availability**
Ground-based observed $PM_{2.5}$ concentrations from CNEMC are available at
https://quotsoft.net/air/ (last access: September 2024). AOD from MODIS Deep Blue retrieval are
available at https://modis.gsfc.nasa.gov/ (last access: September 2024). ERA5 reanalysis data are
available at https://cds.climate.copernicus.eu/ (last access: September 2024). GPM data are
available at
https://disc.gsfc.nasa.gov/datasets/GPM_3IMERGM_07/summary?keywords=%22IMERG%20fi
nal%22 (last access: September 2024). GPCP data are available at
https://psl.noaa.gov/data/gridded/data.gpcp.html (last access: April 2025). The source code of
CESM is available at https://github.com/ESCOMP/CESM (last access: September 2024). Our
model results can be available at https://doi.org/10.5281/zenodo.13682943 (last access: September

492    2024).

**Author Contributions**
Y.Y. conceived the research and directed the analysis. J.G. conceived the research, conducted
the model simulations, and performed the analysis. All the authors including H.W., P.W., and H.
L. discussed the results and wrote the paper.
**Competing Interests**
At least one of the (co-)authors is a member of the editorial board of Atmospheric Chemistry
and Physics. The peer-review process was guided by an independent editor, and the authors also
have no other competing interests to declare.

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

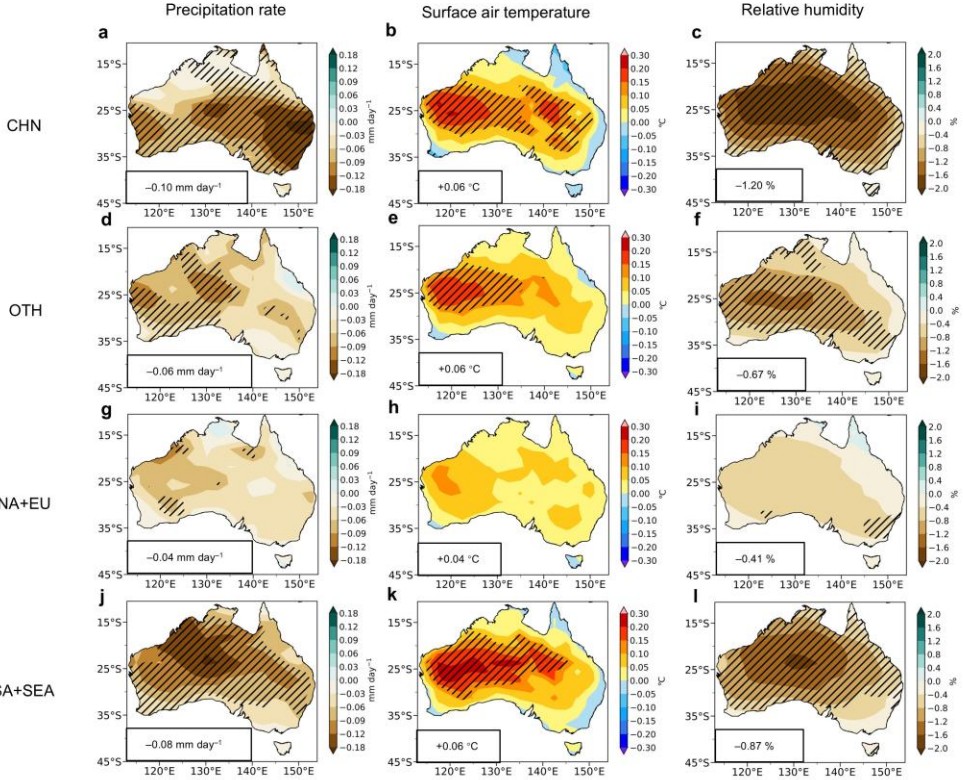


**Figure 1. Simulated changes in precipitation rate, surface air temperature and relative humidity in Australia due to aerosol changes between 2013 and 2019.** Spatial distributions of simulated differences in annual mean precipitation rate (Pr, **a**, **d,** and **g**, unit: mm day$^{-1}$), surface air temperature (TS, **b**, **e**, and **h**, unit: °C) and relative humidity (RH, **c**, **f**, and **i**, unit: %) in Australia between BASE and CHN (CHN minus BASE, **a**–**c**), between BASE and OTH (OTH minus BASE, **d**–**f**), between BASE and NAEU (NAEU minus BASE, **g**–**i**), and between BASE and SASEA (SASEA minus BASE, **j**–**l**). The shaded areas indicate results are statistically significant at the 90% confidence level. Regional averages over Australia are noted at the bottom-left corner of each panel.

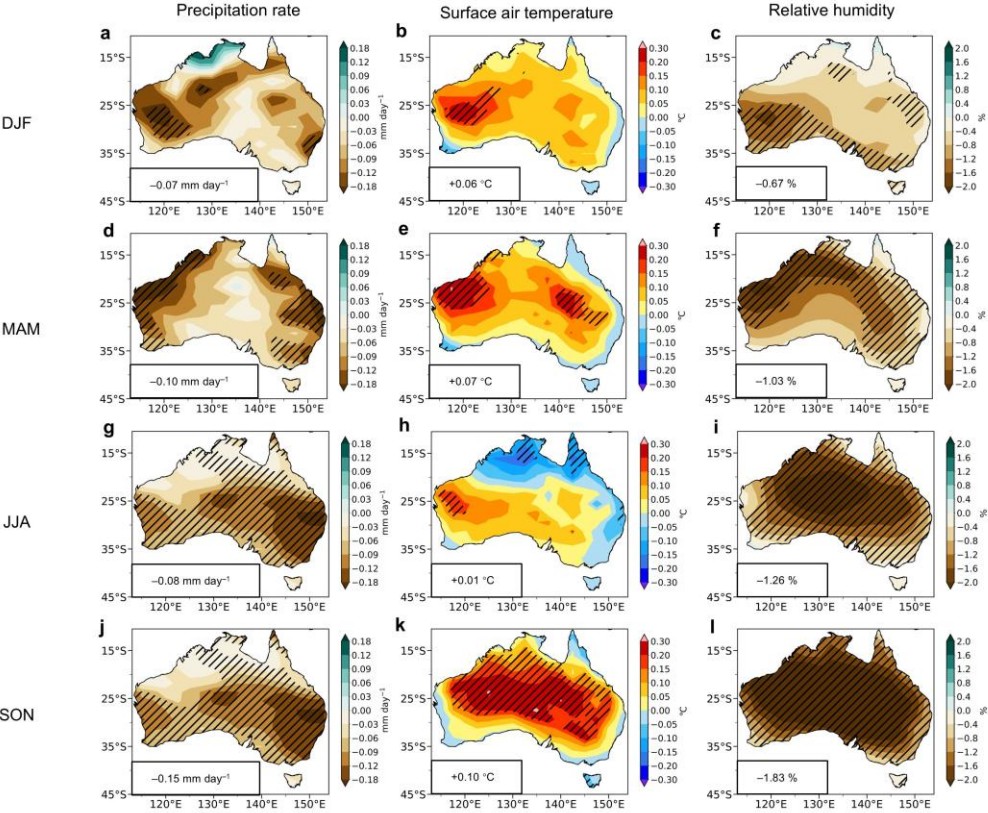

793

**Figure 2. Simulated changes in precipitation rate, surface air temperature and relative humidity in Australia due to aerosol changes in China between 2013 and 2019.** Spatial distributions of simulated differences in DJF (December, January and February, **a**–**c**), MAM (March, April and May, **d**–**f**), JJA (June, July and August, **g**–**i**) and SON (September, October and November, **j**–**l**) mean precipitation rate (Pr, **a**, **d**, **g**, and **j**, unit: mm day$^{-1}$), surface air temperature (TS, **b**, **e**, **h**, and **k**, unit: °C) and relative humidity (RH, **c**, **f**, **i**, and **l**, unit: %) in Australia between BASE and CHN (CHN minus BASE).The shaded areas indicate results are statistically significant at the 90% confidence level. Regional averages over Australia are noted at the bottom-left corner of each panel.

803

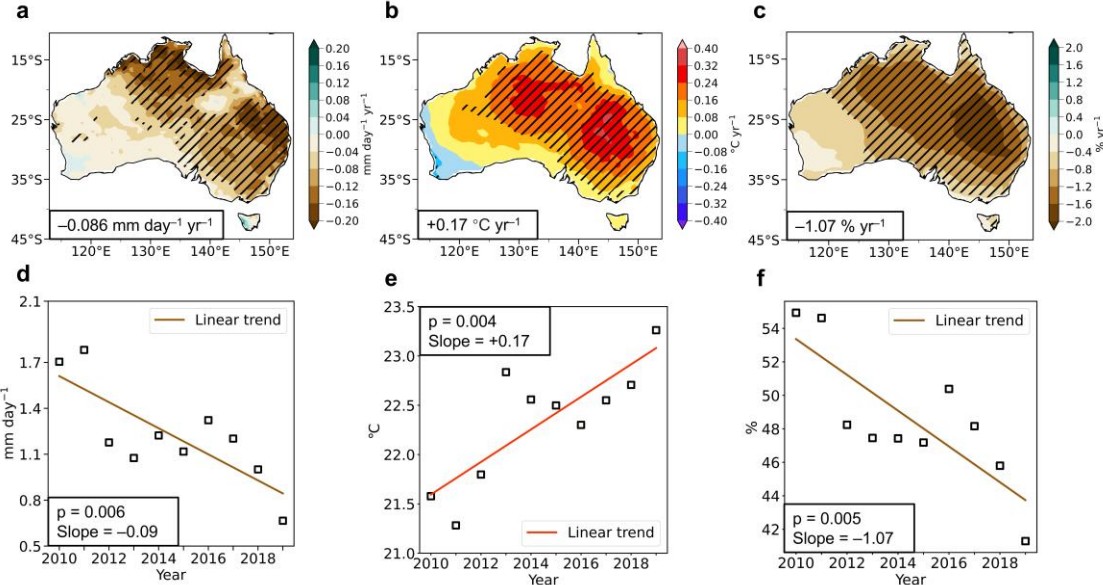

804

**Figure 3. Linear trends of observed precipitation rate, surface air temperature and relative humidity in Australia based on ERA5.** Spatial distributions of linear trends (**a**, **b**, and **c**) and time series (**d**, **e**, and **f**) of annual mean precipitation rate (Pr, **a** and **d**, unit: mm day$^{-1}$), surface air temperature (TS, **b** and **e**, unit: °C) and relative humidity (RH, **c** and **f**, unit: %) in Australia during 2010–2019 from ERA5 reanalysis. The shaded areas indicate trends are statistically significant at the 90% confidence level. Regional averages over Australia are noted at the bottom-left corner of panels a, b, and c. The p values and slopes of linear trends are noted in panels d, e, and f.

812

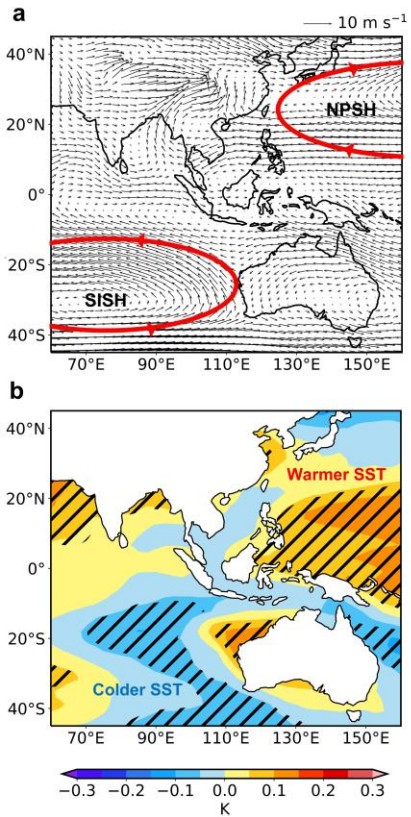

813

**Figure 4. Climatological mean wind fields at 850 hPa and Simulated sea surface temperature changes due to aerosol changes in China between 2013 and 2019. a**, Climatological mean wind fields (unit: m s$^{-1}$, vectors) at 850 hPa from the BASE experiment. NPSH and SISH shown in red circles represents North Pacific Subtropical High and Southern Indian ocean Subtropical High, respectively. **b**, Spatial distributions of simulated differences in annual mean sea surface temperature (SST, unit: mm day$^{-1}$) in Australia between BASE and CHN (CHN minus BASE). The shaded areas indicate results are statistically significant at the 90% confidence level.

821

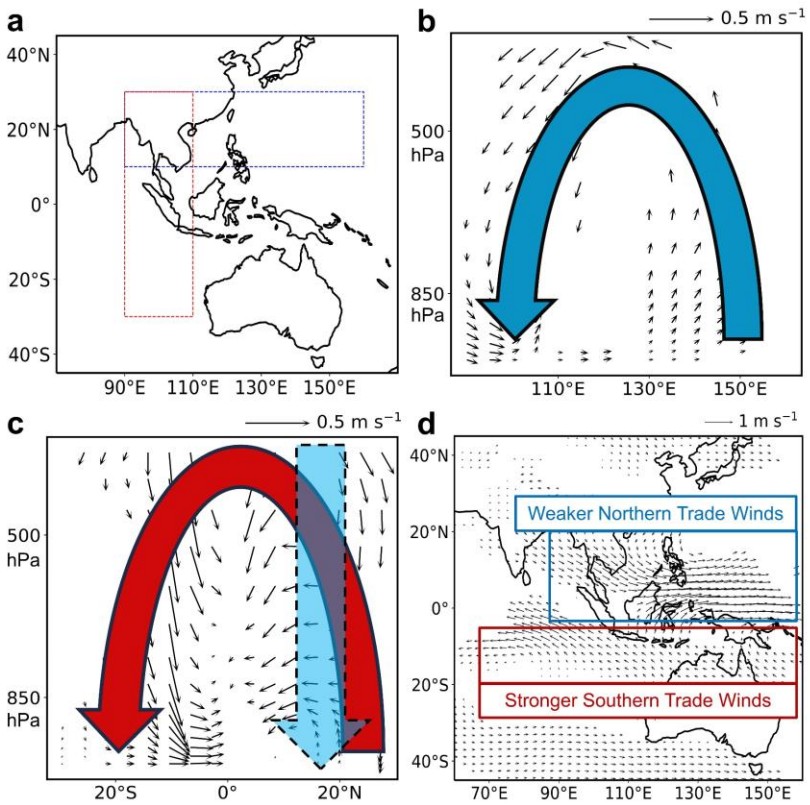

822

**Figure 5. Simulated changes in vertical circulations and 850 hPa wind fields in Asia-Pacific regions due to aerosol changes in China between 2013 and 2019.** Panel **b** and **c** shows pressure–longitude and pressure–latitude cross-section of responses in annual mean atmospheric circulations (unit: m s$^{-1}$, vectors), respectively, over the areas marked with the blue and red box in panel **a**. Panel **d** shows annual mean changes in wind fields (unit: m s$^{-1}$, vectors) at 850 hPa in Asia-Pacific regions. Only atmospheric circulations and winds statistically significant at the 90% confidence level are shown.

830

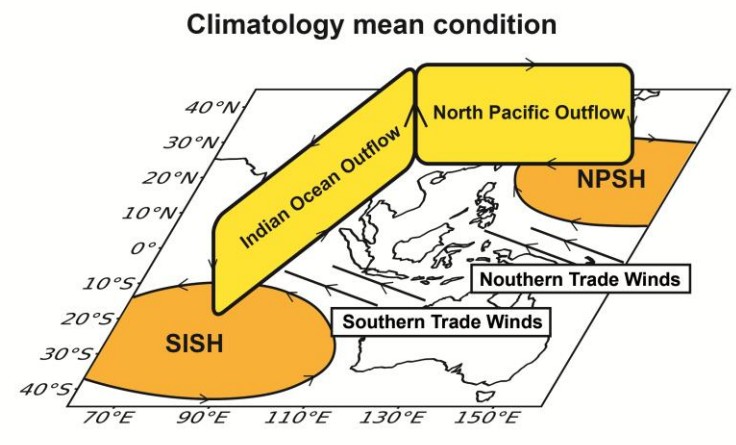

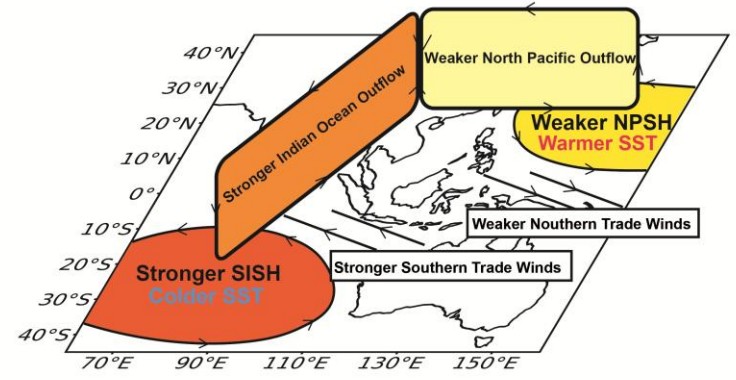

831

**Figure 6. Schematic of the response in large-scale 3D circulations in the Asian-Pacific region to aerosol reductions in China.** The top panel shows climatology mean condition, and the bottom panel shows anomalies resulting from aerosol reductions in China.

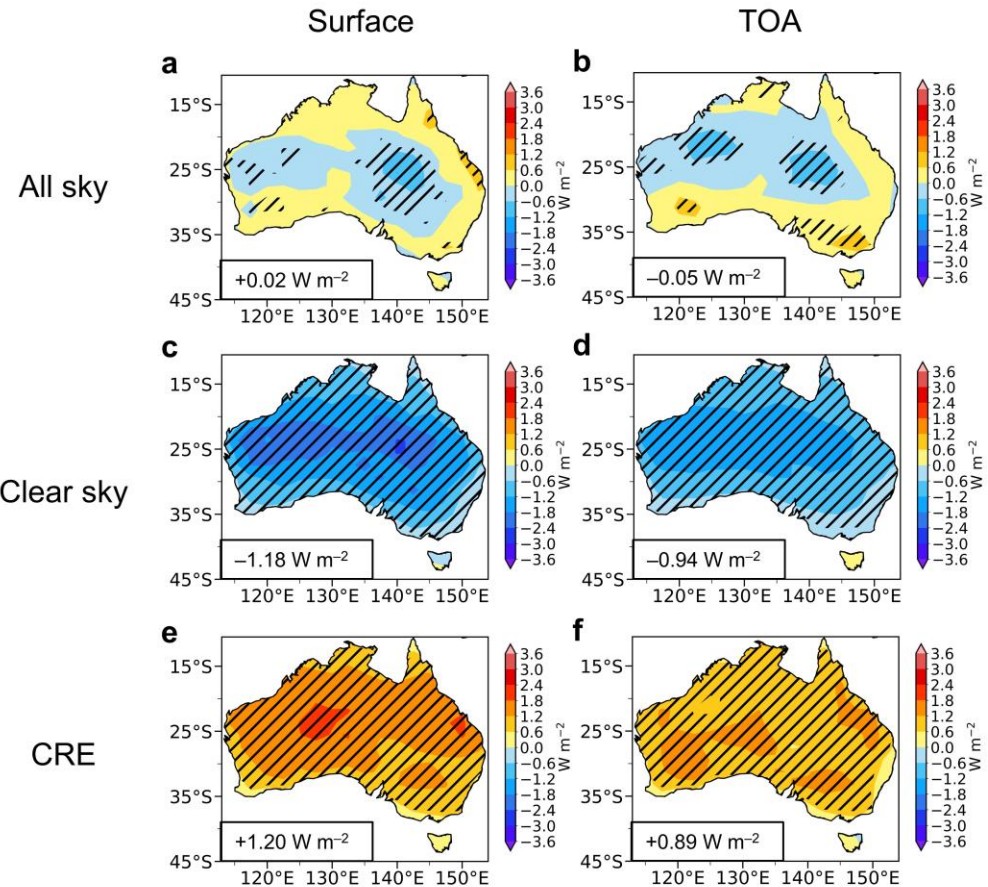

835

**Figure 7. Simulated changes in surface and Top of the Atmosphere (TOA) net total radiative flux under all sky conditions, under clear sky conditions, and from cloud radiative effects in Australia due to aerosol changes in China between 2013 and 2019.** Spatial distributions of simulated differences in annual mean surface (**a**, **c**, and **e**) and Top of the Atmosphere (TOA, **b**, **d**, and **f**) net total radiative flux (unit: W m$^{-2}$) under all sky conditions (**a** and **b**), under clear sky conditions (**c** and **d**), and from cloud radiative effects (CRE, **e** and **f**) in Australia between BASE and CHN (CHN minus BASE). Cloud radiative effects refer to differences under all sky and clear sky conditions (All sky minus Clear sky). The shaded areas indicate results are statistically significant at the 90% confidence level. Regional averages of the responses over Australia are noted at the bottom-left corner of each panel.

846

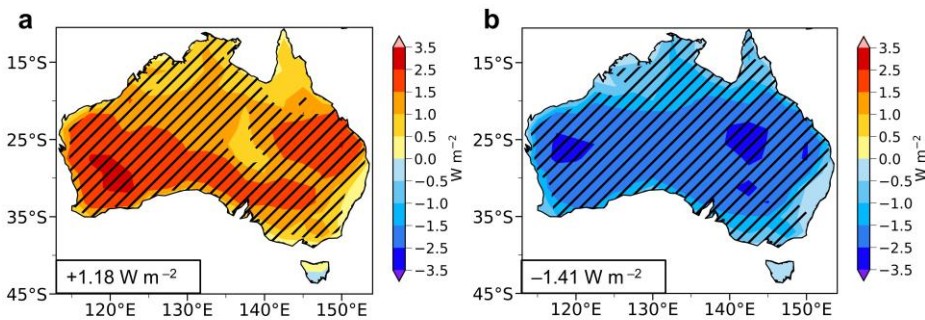

847

**Figure 8. Simulated changes in surface upward sensible and latent heat flux in Australia due to aerosol changes in China between 2013 and 2019.** Spatial distributions of simulated differences in annual mean surface upward sensible (**a**) and latent (**b**) heat flux (unit: W m$^{-2}$) in Australia between BASE and CHN (CHN minus BASE). The shaded areas indicate results are statistically significant at the 90% confidence level. Regional averages over Australia are noted at the bottom-left corner of each panel.

854

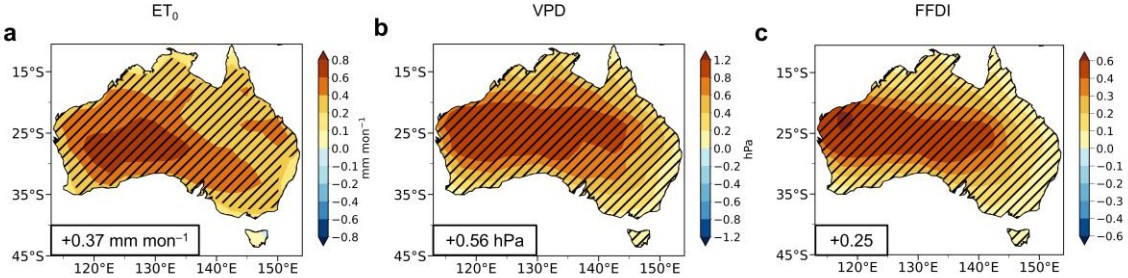

**Figure 9. Simulated changes in reference potential evapotranspiration, vapor pressure deficit, and McArthur forest fire danger index during fire seasons in Australia due to aerosol changes in China between 2013 and 2019.** Spatial distributions of simulated changes in reference potential evapotranspiration (ET$_0$, **a**, unit: mm mon$^{-1}$), vapor pressure deficit (VPD, **b**, unit: hPa), and McArthur forest fire danger index (FFDI, **c**, unitless) during fire seasons (austral spring and summer, from September to the February of the next year) in Australia between BASE and CHN (CHN minus BASE). The shaded areas indicate results are statistically significant at the 90% confidence level. Regional averages over Australia are noted at the bottom-left corner of each panel.