# Peer review of "Dry and warm conditions in Australia exacerbated by aerosol reduction in China Jiyuan Gao1, Yang Yang1\*, Hailong Wang2, Pinya Wang1, Hong Liao1 1Joint International Research Laboratory of Climate and Environment Change (ILCEC), Jiang"

_EGUsphere, 2024_

## Referee Comment (RC1)

**Review of Gao et al - Dry and warm conditions in Australia exacerbated by aerosol reduction in China**

This research article investigates the impact of anthropogenic aerosol reductions in China on Australia's climate. The study found that the decline in Chine's aerosols since 2013 contributed to drier and warmer conditions in Australia by altering temperature and pressure gradients, which intensified the Southern Trade Winds and caused moisture divergence over Australia. The study also links these climate changes to an increase in wildfire risks in Australia. This research highlights the significant influence of distant aerosols on regional climate and offers insights for drought and wildfire risk mitigation.

The manuscript is interesting, well written and tackles an important topic of research (i.e., impact of Chinese aerosols on Australian climate). However, some technical details between the comparison of modelling results and observations need to be corrected and the selection of figures should be adjusted. I recommend acceptance of the manuscript if the major comment below can be addressed.

**Major comments**

1. One of my main comments is related to the comparison of observation/reanalysis data and simulated results: There seems to be some inconsistency between the timeperiods used. In the method section it is mentioned that the period 2013-2019 is used for the observation/reanalysis data as well as the simulated data. However, in the captions of the supplementary figures as well as in the description of these figures in the text (e.g. L274, 278) it is mentioned that the observations are for 2010-2019. Please clarify if the same timeperiod is used for observation/reanalysis data and modelling data and if that is not the case, the plots have to be redone for the correct timeperiod to ensure an accurate comparison. Besides, is this warming and drying trend over Australia still continuing or why did the authors look at the time period 20213-2019?

2. While it is great that the author's tried to reduce the figures in the main text to only 4 to explain the whole story, in particular the mechanistic analysis (Section 3.2) is difficult to follow for the reader with the limited number of figures. For instance, a combination of Figure S16 and S17 (i.e. the filled contours showing the SST pattern overlaid by the climatological wind field) would be a relevant figure to show. Additionally, Figure S21 is heavily referenced in the manuscript but the figure is only shown in the supplementary. Besides, maybe a small schematic of the described mechanism similarly as in Fahrenbach et al. 2024 would be helpful to guide the reader through the description.

3. On the topic of figures, it would be important to show a comparison of the simulated changes with the observed precipitation pattern (Figure S8) as well as the observed wind changes (Figure S18). This is particularly relevant since the authors are trying to do an "attribution" study and it has to be quantified that the observed and modelled changes agree. Additionally, the authors claim that the modelled and observed wind changes are similar (L319-321). While I do acknowledge that 3D wind changes are not the most reliable fields in reanalysis data, this is a bit of an overstatement. Figure S18b and c show very few significant changes making it difficult to understand the simulated flow and Figure S18d shows the largest significant trends in the winds east of Borneo and around

southern Australia, while the authors describe the weaker northern Trade winds and stronger southern Trade winds based on the simulated data. Maybe the authors could think about showing all wind vectors and colouring the significant ones in, so that the reader can at least see if the observations show the same trend even if they might not be significant based on this test?

4. My last comment regarding the figures is that the figure S15 should also be included in the main text. It seems biased to try to find a link / attribution but only show the plots for China which the authors have identified as the relevant one. Maybe a figure showing the annual precipitation trends for CHN, OTH, NA+EU and then a seasonal plot for the CHN plots would be best?

5. The authors discuss the influence of the (very strong) low bias in PM2.5 in CESM1 compared to the observations in L385-388, which is good and relevant. However, this should also be mentioned throughout the manuscript, for instance when the authors try to estimate very precise values for the influence of the Chinese aerosol reductions on precipitation and temperature (L270-271).

**Minor comments**

1. L29-31: The times mentioned in this sentence seem confusing since when first reading it seems that a trend from 2013 is caused by something happening around the 2010s. Maybe using "conditions since the 2010s" would help to settle this confusing sentence.
2. L68-70: Please change "increasing GHGs" to increasing GHG emissions.
3. L72-75: This sentence is very long and confusing, please split it up into two or shorten it
4. L73: "Earth's" instead of "earth's"
5. L104: "especially in northern Australia/especially in the North of Australia" instead of "especially the northern Australia"
6. L104: "affected by the Australian monsoon" instead of "affected by Australian monsoon"
6. L153-159: Is there a reason for the choice of the GPM dataset rather than for instance GPCP data?
7. L224: "Earth's surface" instead of "earth's surface"
8. L244: The setting of DF to 10 according to Sharples et al 2009 needs some more explanation. At least one sentence why Sharples et al choose this value and why it is also applicable here.
9. L277: Please change "evidence" to "indication".
10. L389: Please use "Earth System Model" or "fully-coupled climate model" instead of "aerosol-climate model" which would imply to me that this model is not fully coupled (which is the case according to the method section)
11. Figure S3: The colourbar of these two plots should be the same as the reader might be tricked into thinking that the magnitude changes between the observed and modelled data are similar.

---

## Author Comment (AC1)

**Responses to Referee #1:**

This research article investigates the impact of anthropogenic aerosol reductions in China on Australia's climate. The study found that the decline in Chine's aerosols since 2013 contributed to drier and warmer conditions in Australia by altering temperature and pressure gradients, which intensified the Southern Trade Winds and caused moisture divergence over Australia. The study also links these climate changes to an increase in wildfire risks in Australia. This research highlights the significant influence of distant aerosols on regional climate and offers insights for drought and wildfire risk mitigation.

The manuscript is interesting, well written and tackles an important topic of research (i.e., impact of Chinese aerosols on Australian climate). However, some technical details between the comparison of modelling results and observations need to be corrected and the selection of figures should be adjusted. I recommend acceptance of the manuscript if the major comment below can be addressed.

We thank the reviewer for the constructive suggestions, which are very helpful for improving the clarity and reliability of the manuscript. Please see our point-by-point responses (in blue) to your comments below.

Major comments

One of my main comments is related to the comparison of observation/reanalysis data and simulated results: There seems to be some inconsistency between the timeperiods used. In the method section it is mentioned that the period 2013-2019 is used for the observation/reanalysis data as well as the simulated data. However, in the captions of the supplementary figures as well as in the description of these figures in the text (e.g. L274, 278) it is mentioned that the observations are for 2010-2019. Please clarify if the same timeperiod is used for observation/reanalysis data and modelling data and if that is not the case, the plots have to be redone for the correct timeperiod to ensure an accurate comparison. Besides, is this warming and drying trend over Australia still continuing or why did the authors look at the time period 2013-2019?

The warming and drying trends in Australia due to the reduction of aerosols in China were simulated based on the two simulations with anthropogenic emissions of aerosols and precursors at years 2013 and 2019. It is because China implemented the "Air Pollution Prevention and Control Plan" in 2013 and established nationwide $PM_{2.5}$ observation sites in 2013. Whether the simulated climate responses can be detected in the real world requires the observational evidence. However, the seven years of period 2013–2019 are too short to fully capture the trends in climate variables, as various natural variabilities could influence temperature and rainfall in Australia. For example, climate variabilities, such as ENSO, can have significant impacts on the Australian monsoon and rainfall patterns and influence the long-term trends of temperature and precipitation in Australia. Therefore, we included more years to calculate the trends of the climate variables in observations. It will not affect the results since that the aerosol reductions in China only account a small amount of the changing climate in Australia. We have now added the explanation in the manuscript as "Note that, the trends in observations are calculated during 2010–2019 to minimize the internal variability.". Regarding whether the trend is continuing, CSIRO and BOM (2022) reported that the warming is still ongoing in Australia and the drying trend is

still ongoing in Northern Australia, which could be contributed by the further reductions in Asian aerosols.

While it is great that the author's tried to reduce the figures in the main text to only 4 to explain the whole story, in particular the mechanistic analysis (Section 3.2) is difficult to follow for the reader with the limited number of figures. For instance, a combination of Figure S16 and S17 (i.e. the filled contours showing the SST pattern overlaid by the climatological wind field) would be a relevant figure to show. Additionally, Figure S21 is heavily referenced in the manuscript but the figure is only shown in the supplementary. Besides, maybe a small schematic of the described mechanism similarly as in Fahrenbach et al. 2024 would be helpful to guide the reader through the description.

As requested, we have moved several figures in the main text and added the schematic diagram as shown below.

[Figure]

**Figure 6. Schematic of the response in large-scale 3D circulations in the Asian-Pacific region to aerosol reductions in China.** The top panel shows climatology mean condition, and the bottom panel shows anomalies resulting from aerosol reductions in China.

On the topic of figures, it would be important to show a comparison of the simulated changes with the observed precipitation pattern (Figure S8) as well as the observed wind changes (Figure S18). This is particularly relevant since the authors are trying to do an "attribution" study and it has to be quantified that the observed and modelled changes agree. Additionally, the authors claim that the modelled and observed wind changes are similar (L319-321). While I do acknowledge that 3D wind changes are not the most reliable fields in reanalysis data, this is a bit of an overstatement. Figure S18b and c show very few significant changes making it difficult to understand the simulated flow and Figure S18d shows the largest significant trends in the winds east of Borneo and around stronger southern Trade winds based on the simulated data.

Maybe the authors could think about showing all wind vectors and colouring the significant ones in, so that the reader can at least see if the observations show the same trend even if they might not be significant based on this test?

Thank you for your suggestion. We have revised the figure displaying the 3D wind fields, with significant and insignificant circulations clearly distinguished. Additionally, the statement has been revised to: "Although only a few significant changes persist in Northern Australia, the large-scale circulations around Australia show noticeable similarities to the simulated results." to avoid the overstatement.

[Figure]

**Figure S17. Linear trends in observed vertical circulations and 850 hPa wind fields in Asia-Pacific regions.** Panel **b** and **c** shows pressure–longitude and latitude cross-section of linear trends in annual mean atmospheric circulations (unit: m s$^{-1}$, vectors) over areas marked with the blue and red box in panel **a** during 2010–2019 from ERA5. Panel **d** shows linear trends of wind fields (unit: m s$^{-1}$, vectors) at 850 hPa in Asia-Pacific regions. Trends of atmospheric circulations and winds which are statistically significant at the 90% confidence level are shown in black, while the insignificant ones are shown in grey.

My last comment regarding the figures is that the figure S15 should also be included in the main text. It seems biased to try to find a link / attribution but only show the plots for China which the authors have identified as the relevant one. Maybe a figure showing the annual precipitation trends for CHN, OTH, NA+EU and then a seasonal plot for the CHN plots would be best?

As requested, we have moved the figure in the main text and showed the CHN, OTH, NA+EU (Figure 1 below) and then a seasonal plot for the CHN (Figure 2 below).

[Figure]

**Figure 1. Simulated changes in precipitation rate, surface air temperature and relative humidity in Australia due to aerosol changes between 2013 and 2019.** Spatial distributions of simulated differences in annual mean precipitation rate (Pr, **a**, **d,** and **g**, unit: mm day$^{-1}$), surface air temperature (TS, **b**, **e**, and **h**, unit: °C) and relative humidity (RH, **c**, **f**, and **i**, unit: %) in Australia between BASE and CHN (CHN minus BASE, **a**–**c**), between BASE and OTH (OTH minus BASE, **d**–**f**), and between BASE and NAEU (NAEU minus BASE, **g**–**i**). The shaded areas indicate results are statistically significant at the 90% confidence level. Regional averages over Australia are noted at the bottom-left corner of each panel.

[Figure]

**Figure 2. Simulated changes in precipitation rate, surface air temperature and relative humidity in Australia due to aerosol changes in China between 2013 and 2019.** Spatial distributions of simulated differences in DJF (December, January and

February, **a**–**c**), MAM (March, April and May, **d**–**f**), JJA (June, July and August, **g**–**i**) and SON (September, October and November, **j**–**l**) mean precipitation rate (Pr, **a**, **d**, **g**, and **j**, unit: mm day$^{-1}$), surface air temperature (TS, **b**, **e**, **h**, and **k**, unit: °C) and relative humidity (RH, **c**, **f**, **i**, and **l**, unit: %) in Australia between BASE and CHN (CHN minus BASE).The shaded areas indicate results are statistically significant at the 90% confidence level. Regional averages over Australia are noted at the bottom-left corner of each panel.

The authors discuss the influence of the (very strong) low bias in PM2.5 in CESM1 compared to the observations in L385-388, which is good and relevant. However, this should also be mentioned throughout the manuscript, for instance when the authors try to estimate very precise values for the influence of the Chinese aerosol reductions on precipitation and temperature (L270-271).

To address this, we have now included notes regarding this low bias and its potential impact on the accuracy of these estimates in the relevant sections throughout the manuscript, such as: "However, considering that aerosols are underestimated in the model, the impact of aerosol reductions in China on Australia's climate/wildfire risks may also be underestimated."

Minor comments

L29-31: The times mentioned in this sentence seem confusing since when first reading it seems that a trend from 2013 is caused by something happening around the 2010s. Maybe using "conditions since the 2010s" would help to settle this confusing sentence.

Revised.

L68-70: Please change "increasing GHGs" to increasing GHG emissions.

Changed.

L72-75: This sentence is very long and confusing, please split it up into two or shorten it

This sentence has been shortened as: "Atmospheric aerosols are the second-largest anthropogenic climate forcer, exerting an overall cooling effect that partially masks the warming induced by GHGs."

L73: "Earth's" instead of "earth's"

OK.

L104: "especially in northern Australia/especially in the North of Australia" instead of "especially the northern Australia"

Revised.

L104: "affected by the Australian monsoon" instead of "affected by Australian monsoon"

Revised.

L153-159: Is there a reason for the choice of the GPM dataset rather than for instance GPCP data?

"GPM provides higher temporal and spatial resolution data compared to GPCP, making it more suitable for studies focused on short-term precipitation variability and regional climate dynamics." The statement has been added to the manuscript.

L224: "Earth's surface" instead of "earth's surface"

Revised.

L244: The setting of DF to 10 according to Sharples et al 2009 needs some more explanation. At least one sentence why Sharples et al choose this value and why it is also applicable here.

Thank you for your suggestion. Sharples et al. (2009) mentioned that "Such a factor will have no real bearing on the methods of comparison employed in the later sections of the paper and so for convenience we assume that DF = 10 in what follows." We acknowledge that this assumption is idealized. To address this, we have calculated gridded DFs (Figure S7 below) and found that the DFs in Australia are close to 10, with spatial distributions being nearly homogeneous. Therefore, setting DF = 10 for Australia is reasonable in this context.

[Figure]

**Figure S7. Spatial distributions of annual mean dry factor (unit: unitless) in Australia during 2010–2019.** The data is from fire danger indices historical data from the Copernicus Emergency Management Service (CEMS, 2019; Vitolo et al., 2020).

Additionally, we compare FFDI with DF = 10 and FFDI with the gridded DF (Figure S8 below). The patterns and regional averages of both are similar.

[Figure]

**Figure S8. Spatial distribution of simulated changes in FFDI (unit: unitless) during fire seasons in Australia between BASE and CHN (CHN minus BASE).** Shaded areas indicate results that are statistically significant at the 90% confidence level. Regional averages for Australia are noted at the bottom-left corner of each panel. The left panel shows FFDI (DF = 10), and the right panel shows FFDI (gridded DF).

L277: Please change "evidence" to "indication".

Changed.

L389: Please use "Earth System Model" or "fully-coupled climate model" instead of "aerosol-climate model" which would imply to me that this model is not fully coupled (which is the case according to the method section)

Thanks for your reminder. The term "aerosol-climate model" has been replaced with "fully-coupled climate model" throughout the manuscript, as suggested.

Figure S3: The colourbar of these two plots should be the same as the reader might be tricked into thinking that the magnitude changes between the observed and modelled data are similar.

Thank you for your suggestion. If we use the same color scale for both plots, the color range of the modeled results becomes overly uniform. However, we have now added a note in the figure caption to remind the reader that the magnitudes of the observed and modeled data are not directly comparable, and that the color scales represent different ranges.

**References:**

CSIRO and BOM: State of the Climate 2022, 2022

CEMS: Fire danger indices historical data from the Copernicus Emergency Management Service, https://doi.org/10.24381/cds.0e89c522, 2019.

Vitolo, C., Di Giuseppe, F., Barnard, C., Coughlan, R., San-Miguel-Ayanz, J., Libertá, G., and Krzeminski, B.: ERA5-based global meteorological wildfire danger maps, Sci. Data, 7, 216, https://doi.org/10.1038/s41597-020-0554-z, 2020.

---

## Author Comment (AC2)

**Responses to Referee #2:**

The manuscript investigates the role of the recent decline in East Asian anthropogenic aerosols in driving precipitation and temperature changes across Australia. The authors highlight the potential for aerosols to contribute to enhancing the fire risk via widespread drying and warming. The topic is timely and very important, as not many studies have examined the influence of recent changes in aerosols on recent climate variations. However, I find several major weaknesses in the study, including in the simulation design, that prevent the manuscript from being acceptable for ACPD. Unfortunately, I need to recommend a rejection. Happy though to reconsider the manuscript if reviewed accordingly.

Thank you for your feedback. We will carefully consider the points you raised regarding the simulation design and other aspects of the study. We value your insights and will make every effort to improve the quality of our work. We would be grateful for your reconsideration of the manuscript. Please see our point-by-point responses (in blue) to your comments below.

Major comments:

I believe there is a serious overinflation of the recent observed rainfall changes over Australia. Firstly, the authors use observations and plot linear trends from 2010, while model results are shown as 3-year differences from 2013. While the latter accounts for the decline in aerosols, observations show that the years 2010 and 2011 were anomalously wet over Australia, and just a couple of years before rainfall was much less (I have checked by plotting observations since 2000). This can also be inferred by examining Fig 2, where clearly the trend in panel d is affected by the two early years. By eye, using 3-year composites from 2013, the rainfall changes are very modest. Looking at more recent years, it turns out that 2019 was also anomalously dry, and more recent years (excluding those affected by COVID-related reduced emissions) show a recovery. Therefore the results based on observations, which ultimately is the motivation of the study, are strongly affected by the extremely short record and cannot be trusted.

Thank you for pointing out the strong variability in rainfall in Australia. We agree with reviewer that the observed trend of rainfall could be affected by the wet years in 2010/2011 and dry year in 2019, which could be induced by the interval variability (such as ENSO). However, we could not deny the decreasing precipitation during 2010–2019 and rule out the potential influence of changing aerosols on the rainfall variation. Additionally, we are trying to quantify the role of aerosol reduction in China in the Australian climate, rather than fully attribute to changing Australian climate to aerosols in China, considering that aerosol reductions in China are found to contribute to small part of the changing precipitation in Australia. The simulation results are still robust enough to support our conclusions because our model experiments are based on validated climate model, providing strong evidence that aerosol reductions in China have a significant impact on Australia's climate.

In fact, the emission reduction of aerosols and precursors in China starts in 2010 rather than 2013. We chose the year 2013 in simulation was because China implemented the "Air Pollution Prevention and Control Plan" in 2013 and established nationwide $PM_{2.5}$ observation sites in 2013. As the reviewer mentioned, Australia

experienced a transient from dry years before 2010 to wet years in 2010/2011 and the rainfall recovery after 2020. It is also in accordance with the increasing aerosols in China before 2010 and the slowdown in aerosol reduction in recent years. But it still requires quantitative analysis.

Related to the point above, it is not surprising that observed circulation trends (e.g., Fig. S18) are extremely weak. Yet, this is the mechanism driving Australian rainfall changes, thus is central to the proposed aerosol influence.

As mentioned earlier, given natural variability and the multiple factors influencing precipitation in Australia, it is quite challenging to capture patterns in the observations that resemble the results of the sensitivity experiment on China's aerosol reductions. We have redrawn the circulation figure (Figure S17 below), displaying all circulations regardless of whether they pass the significance test. Notably, we can observe clear similarities with the model results.

[Figure]

**Figure S17. Linear trends in observed vertical circulations and 850 hPa wind fields in Asia-Pacific regions.** Panel **b** and **c** shows pressure–longitude and latitude cross-section of linear trends in annual mean atmospheric circulations (unit: m s$^{-1}$, vectors) over areas marked with the blue and red box in panel **a** during 2010–2019 from ERA5. Panel **d** shows linear trends of wind fields (unit: m s$^{-1}$, vectors) at 850 hPa in Asia-Pacific regions. Trends of atmospheric circulations and winds which are statistically significant at the 90% confidence level are shown in black, while the insignificant ones are shown in grey.

The model analysis is also weak in the sense that 3-year composites are extremely short to identify forced trends. In addition, observed trends should be first compared to those from a control simulation and the contribution of aerosols should be related to the model world, not observations.

We believe there may be a misunderstanding regarding the 3-year composites mentioned by the reviewer. In our study, we did not use 3-year composites for climate

variables (only for PM$_{2.5}$ and AOD in observations since they are more linear than climate variables). Our equilibrium experiment designs have been described in the 2.2 Model Description and Experimental Design section.

Regarding the model comparison, we would like to clarify that due to the nature of our equilibrium experiments, direct comparison with baseline control experiments is not feasible. Even in transient simulations, models may not be able to fully capture the short-term and complex observational signal, as observational data are influenced by numerous factors over short time periods. However, models can provide a quantitative evaluation of how specific factors, such as aerosols, might contribute to climate phenomenon. This approach has been commonly used in previous studies (Bollasina et al., 2011; Heede and Fedorov, 2021; Hwang et al., 2024; Vecchi et al., 2006; Wang et al., 2023; Yang et al., 2022) to isolate the impact of individual forcers. This limitation has been noted in the discussion section: "Another limitation of this study is that we calculated the relative contribution of aerosol changes in China to the changing climate in Australia by combining model simulations and observational data, which could lead to the inconsistency and introduce biases to the quantitative results.".

Another major weakness is the experimental design. The authors examine long-term changes coming from equilibrium runs, which are far from representing the actual transient response over 10 years or so. This is likely far from being in equilibrium. I would have seen large-ensemble transient simulations with time-varying aerosols (2013-2024 for example) as more appropriate and suitable.

We appreciate the reviewer's concern regarding the use of equilibrium runs and the suggestion to use large-ensemble transient simulations. While transient simulations offer a more dynamic representation of temporal changes, the relatively short period from 2013 to the present may not provide enough time for the climate system to fully respond to aerosol changes. This is particularly true when considering the long timescales required for oceanic heat capacity to equilibrium. Furthermore, equilibrium simulations allow for a clearer isolation of the forced response to aerosol changes, minimizing the impact of short-term climate variability.

We do acknowledge, however, that transient simulations would offer a more accurate depiction of the time-varying effects of aerosol reductions, especially as aerosol reductions in China continue. As the climate system responds more robustly over time, transient simulations will likely become a more appropriate tool. Therefore, while equilibrium simulations are potentially more suitable for the present study, we plan to use transient simulations in future research.

We have also mentioned the limitation of not using transient simulations in the discussion section: "While this study relies on equilibrium simulations to isolate the forced response to aerosol changes, we acknowledge that transient simulations, which account for time-varying aerosol reductions, would provide a more accurate depiction of the dynamic climate system response. Given that aerosol reductions in China continue to evolve, transient simulations would be more appropriate for capturing the full temporal effects of these changes. Large-ensemble transient simulations are suggested in future studies to better represent the evolving aerosol-climate interactions over time and to enhance the robustness of the findings.".

Trends or changes should always be plotted with the related statistical significance, such as in Figs. 1 and 2.

Thank you for the suggestion. Now all trends or changes have been plotted with statistical significance.

Fig 3: I think it would be more appropriate to display the divergent wind, rather than the total wind.

We chose to display the total wind in Figure 3 to highlight the strengthening of Southern Trade Winds. For the divergent winds, we also displayed moisture divergence (Figure S18), which is more directly related to the changes in precipitation patterns that we are investigating.

**References:**

Bollasina, M. A., Ming, Y., and Ramaswamy, V.: Anthropogenic Aerosols and the Weakening of the South Asian Summer Monsoon, Science, 334, 502–505, https://doi.org/10.1126/science.1204994, 2011.

Heede, U. K. and Fedorov, A. V.: Eastern equatorial Pacific warming delayed by aerosols and thermostat response to $CO_2$ increase, Nat. Clim. Chang., 11, 696–703, https://doi.org/10.1038/s41558-021-01101-x, 2021.

Hwang, Y.-T., Xie, S.-P., Chen, P.-J., Tseng, H.-Y., and Deser, C.: Contribution of anthropogenic aerosols to persistent La Niña-like conditions in the early 21st century, Proc. Natl. Acad. Sci. U. S. A., 121, e2315124121, https://doi.org/10.1073/pnas.2315124121, 2024.

Vecchi, G. A., Soden, B. J., Wittenberg, A. T., Held, I. M., Leetmaa, A., and Harrison, M. J.: Weakening of tropical Pacific atmospheric circulation due to anthropogenic forcing, Nature, 441, 73–76, https://doi.org/10.1038/nature04744, 2006.

Wang, P., Yang, Y., Xue, D., Ren, L., Tang, J., Leung, L. R., and Liao, H.: Aerosols overtake greenhouse gases causing a warmer climate and more weather extremes toward carbon neutrality, Nat. Commun., 14, 7257, https://doi.org/10.1038/s41467-023-42891-2, 2023.

Yang, Y., Ren, L., Wu, M., Wang, H., Song, F., Leung, L. R., Hao, X., Li, J., Chen, L., Li, H., Zeng, L., Zhou, Y., Wang, P., Liao, H., Wang, J., and Zhou, Z.-Q.: Abrupt emissions reductions during COVID-19 contributed to record summer rainfall in China, Nat. Commun., 13, 959, https://doi.org/10.1038/s41467-022-28537-9, 2022.

---

## Referee Report (RR1)

**Review Gao et al**

The manuscript now includes more relevant figures in the main and supplementary material. However, the manuscript still has some major caveats related to the experimental setup and observational comparison after the comments from the reviewers and editor. I recommend another round of major revision based on the comments below. If the authors would not be able to give a scientific reasoning why a comparison of the equilibrium simulations with the transient (fast) observed precipitation response over Australia (see major comment 1) is valid, I would recommend that the article is rejected but could be considered for resubmission if the author's performed simulations to assess the fast climate response and find that these simulations justify their hypothesis.

Major comments

1. My main concern is still regarding the experimental setup since the changes over Australia in an equilibrium climate simulation (around 100 years) are compared to the fast climate response in observations (less than 10 years). A paper by Liu et al 2018 examined the fast and slow precipitation response over Australia shows that while the fast response shows a drying to Asian sulfate aerosols, the slow response shows a wettening (see Figure 1). Additionally, a recent paper by Hwang et al 2024 shows very different east-west Pacific and Indian Ocean SST patterns which lead to differences in the flow towards Australia (see Figure 3). Based on this previous literature, it does not seem justified to attribute the recent short-term drying (fast response) in Australia using equilibrium simulations which only show the slow responses.

   In response to the reviewer and editor the authors write the following:
   o „While transient simulations offer a more dynamic representation of temporal changes, the relatively short period from 2013 to the present may not provide enough time for the climate system to fully respond to aerosol changes."
   o "As the climate system responds more robustly over time, transient simulations will likely become a more appropriate tool."

   I would argue that both these statements are not correct: In the former, if the authors argue that the "real world climate" did not have enough time to fully respond to the aerosol changes then the same reasoning should be applied to the modelled climate. Thus, it seems incorrect to compare equilibrium climate simulations where the climate system had a lot of time to fully respond to the aerosol changes with the transient "real world climate". Similarly, the second comment is incorrect as over time the equilibrium simulations will become more appropriate (e.g. if it is a long-term Australian drying trend that should be attributed) while the short-term trend that the authors try to assess would be more accurately captured by a fast response.

   The authors now added a short paragraph in the discussion (L432-439). However, the authors will have to add a detailed discussion of their results (and choice to use an equilibrium simulations) in the light of the papers by Hwang et al 2024 and Liu et al 2018 as well as any other relevant papers. How can this attribution of the fast Australian drying based on equilibrium simulations be trusted if previous literature

shows large differences in the precipitation patterns over Australia and SST around Australia (and related mechanisms) in the fast and slow response? If the authors would be unable to give a scientific reasoning, I would recommend that the article is rejected but could be considered for resubmission if the author's performed simulations to assess the fast climate response and find that these simulations justify their hypothesis.

2. Thanks for providing the additional Figures S12 and S14 which help to examine the effect of different datasets and time periods. However, the authors still focus on the observational 2010-2019 and do not address how the anomalously wet year in 2010/11 might bias their assessment. This is particularly relevant since the authors theoretically want to compare the influence of Chinese aerosols on precipitation trends over Australia from 2013-2019.

   In order to assess the impact of including these three additional years, I recommend to create a spatial precipitation trend figure based on observational data showing the 2013-2019 trend in comparison to the 2010-2019 trend that the authors already show. If the 2013-2019 trend plot shows similar changes as the 2010-2019 plot, then this could help to make their statement of including the additional 3 years to reduce the influence of internal variability more robust.
   Additionally, the large impact of internal variability in the observational data should be discussed in the discussion further.

Liu, L., et al. (2018): "A PDRMIP multimodel study on the impacts of regional aerosol forcings on global and regional precipitation." *Journal of climate* 31.11, 4429-4447.

Hwang, Yen-Ting, et al. (2024): "Contribution of anthropogenic aerosols to persistent La Niña-like conditions in the early 21st century." *Proceedings of the National Academy of Sciences* 121.5

---

## Author Response (AR2)

**Responses to Editor:**

Both reviewers raised major concerns with this manuscript, which I don't think have been sufficiently addressed in either the response or the revisions. I share the reviewers' concerns about the experiment design used in this study, and would like to see these addressed in more detail before returning the manuscript to the reviewers.

Thank you for your comments. We appreciate your feedback. We have carefully revised the manuscript and addressed these concerns in detail. Please find our point-by-point responses below (in blue).

The observed precipitation trend is evaluated based on only 10 years of data, and there is no discussion of how this trend compares to either longer term trends, or internal variability. The trend is also based on only one observational dataset.

• I think it is important that we see a comparison between the GPM data and other precipitation datasets such as GPCP to give a sense of how robust this observed trend is.

We acknowledge the concern regarding the reliance on a single observational dataset for evaluating precipitation trends. To address this, we have compared the GPM dataset with ERA5 and GPCP precipitation data (Fig. S14). Our analysis shows that all three datasets exhibit a similar decreasing trend in precipitation over Australia, reinforcing the robustness of the observed trend. We have incorporated this comparison into the revised manuscript (Results section): "The GPM and GPCP data also exhibit a similar decreasing trend in precipitation over Australia, reinforcing the robustness of the observed drying trend (Figure S14)."

[Figure]

**Figure S14. Linear trends of observed precipitation rate in Australia based on ERA5, GPM, and GPCP.** Spatial distributions of linear trends annual mean precipitation rate (unit: mm day$^{-1}$) in Australia during 2010–2019 from ERA5 (a), GPM (b), and GPCP (c) datasets. The shaded areas indicate trends are statistically significant at the 90% confidence level. Regional averages over Australia are noted at the bottom-left corner of panels.

• It is also important that the authors show a longer precipitation timeseries, and evaluate the interannual variability, to give some context to the magnitude and significance of the trend. Reviewer 2 is concerned that 2010 and 2011 were anomalously wet, and thus bias the trend. They have plotted data from 2000 – I ask the authors to at least do the same, and would strongly encourage them to plot an even longer timeseries to provide more context. It is stated in the abstract that Australia experienced anomalously dry and warm conditions since the 2010s, but, beyond a few citations, there isn't any evidence in the manuscript to support this.

Thank you for your suggestion. As per your request, we have extended the time series of
precipitation covering 2001–2019 in Fig. S12. The updated figure clearly shows an increasing
precipitation trend during 2001–2010, followed by a decreasing trend during 2010–2019.

[Figure]

**Figure S12. Linear trends of observed precipitation rate during 2001–2019 based on ERA5.**
The time series of annual mean precipitation rate (unit: mm day$^{-1}$) over Australia during 2001–
from ERA5 reanalysis. The linear trends for the periods 2001–2010 and 2010–2019 are
indicated. The time series are also given after removing the influence of Niño 3.4 index.

[Figure]

**Figure S13. Scatter plot showing the correlation between SST anomaly in Niño 3.4 region
and precipitation in Australia.** The red line represents the linear regression fit to the data, with
the corresponding correlation coefficient (R) and p-value displayed in the figure.

• In the response document, the authors state that the observed trend is calculated from 2010-2019
instead of 2013-2019 to minimize the internal variability. This is still a very short time period, and
a quantitative assessment of internal variability is required if readers are to be confident that it is a
forced trend.

To account for the potential influence of internal variability, particularly ENSO, which is a major
driver of Australia's precipitation variability (Fig. S13), we have also adjusted the precipitation
data by removing the influence of Niño 3.4 index. The adjusted results exhibit a similar transition, confirming the robustness of the observed forced drying trend since the 2010s (Fig. S12).

The reviewers also raise important concerns about the experiment design, which warrant more
discussion than they have been given in the current response and revision. I am not convinced by
the response to reviewer 2, and think the authors undermine their own interpretation of their results
in this response. Equilibrium runs are an excellent tool for understanding the physical response to
forcing, and for identifying the mechanisms for particular responses, as the authors have done in
this study. However, care needs to be taken when comparing them to real-world changes. As the
authors state in their response, 'the relatively short period from 2013 to the present may not provide
enough time for the climate system to fully respond to aerosol changes'. But, this full response is
exactly what their equilibrium simulations show. So, how can these experiments be used to
quantify an aerosol contribution to an observed trend, as the authors do in their manuscript? Is it
possible to use such simulations to make anything more than qualitative statements about observed
trends? One could perhaps quantify the Chinese aerosol contribution to a modelled response, but
this would require other forcings to be considered. Reviewer 2 is recommending rejection of the
manuscript based on the experiment design. I would like to see a more detailed discussion of the
design of the study, and the implications of comparing several hundred years of simulated
equilibrium responses to just 10 years of observed data, in both the response to the reviewers and
the revised manuscript before making a decision.

Thank you for your insightful comments regarding the experiment design and the comparison
between equilibrium simulations and observed trends. We acknowledge the limitations of our
approach and have carefully revised the manuscript accordingly. The equilibrium simulations
primarily capture the long-term climate response to aerosol forcing, whereas the observed trends
over a decade are influenced by both forced responses and internal variability. While our
simulations provide valuable insights into the physical mechanisms linking aerosol reductions to
climate changes, they are not intended for direct quantitative attribution of observed trends. **To
clarify it, we have removed all statements that directly attribute the observed trend to the
simulated aerosol response in a quantitative manner and now emphasize that the simulations
serve as a tool for understanding the underlying processes rather than quantitatively
estimating real-world aerosol-induced changes.** Moving forward, we recognize that transient
simulations would be better suited for capturing short-term forced trends, and we plan to
incorporate them in future studies to further investigate the evolution of short-term responses and
assess the potential contributions of other external forcings, such as greenhouse gases and internal
variability.

I am confused by the interpretation of the low bias in the PM2.5 and AOD trends in CESM1 found
in the response and the revised manuscript. CESM1 is known to underestimate PM2.5. The authors
have calculated a change in PM2.5 and found that it is small compared to the observed change. If
this is based on an absolute change, and the background PM2.5 in the model is low, then one would
also expect the change to be small compared to observations. However, even though aerosol
concentrations and AOD in the model are low, CESM1 has a relatively large aerosol forcing. Thus,
a small absolute change in PM2.5 or AOD may result in a large forcing in the model. I don't think
a small absolute change in the model necessarily implies an underestimate of the forced response.
How do the percentage changes in PM2.5 and AOD in the model compare to the percentage
changes in observations, for example?

Thank you for your valuable comment. We acknowledge the known low bias in $PM_{2.5}$ and AOD
in CESM1 and appreciate the need for a more nuanced interpretation of the model's response. In our original analysis, we primarily compared absolute changes, which may not fully capture the
model's sensitivity to aerosol forcing given its relatively strong aerosol-cloud interactions. To
address this, we have now included a comparison of percentage changes in PM$_{2.5}$ and AOD
between the model and observations (Fig. S4 and Fig. S5). This allows for a more direct assessment
of whether the model underestimates the forced response, independent of its background aerosol
concentrations. We do note that the underestimation of relative concentrations is much smaller.
The revised discussion clarifies that although CESM1 underestimates aerosol concentrations, the
strong climate sensitivity of the model to aerosol forcing suggests that the simulated climate
response may not be significantly underestimated.

[Figure]

**Figure S4. Comparisons of relative changes in near-surface PM$_{2.5}$ concentrations between observation and model simulation.** Spatial distributions of observed (circles) and modeled (shades) annual mean relative changes (2017–2019 minus 2013–2015, relative to 2013–2015) in near-surface PM$_{2.5}$ concentration (unit: %). Relative changes of observation and model simulation and correlation coefficient (R) between observation and simulation are shown at the bottom-left corner of the panel.

[Figure]

**Figure S5. Comparisons of relative changes in aerosol optical depth (AOD) between satellite retrieval and model simulation.** Spatial distributions of annual mean relative changes (2017–2019 minus 2013–2015, relative to 2013–2015) in Moderate Resolution Imaging Spectroradiometer (MODIS) (**a**) and modeled (**b**) AOD (unit: %). Relative changes of observation and model simulation and correlation coefficient (R) between observation and simulation are shown at the bottom-left corner of panel a.

Thinking back to the discussion of precipitation, it would also be good to know more about the significance of the observed trend. The emission reductions have been very large, but how significant is the change in AOD? Is there potentially a contribution from internal variability here that accounts for some of the model underestimate of the change relative to observations?

Thank you for your thoughtful comment. The observed AOD trend can indeed be affected by factors beyond anthropogenic aerosol changes, such as meteorological variability and natural aerosol sources. However, Zhai et al. (2019) estimated that only 12% of the $PM_{2.5}$ decrease in China during 2013–2018 was attributable to meteorology. Given the substantial magnitude of emission reductions, the influence of these other factors is relatively minor, and the observed AOD trend remains statistically significant over large regions of China (Fig. R1). This significant decreasing AOD trends in observations have also been reported in many previous studies. Therefore, we don't think the modeled AOD bias could be largely attributed to the internal variability in observations. We have revised the manuscript to clarify these points.

[Figure]

**Figure R1. Linear trends of observed AOD in Australia based on MODIS.** Spatial distributions of linear trends of annual mean AOD (unit: unitless) in China during 2010–2019 from MODIS. The shaded areas indicate trends are statistically significant at the 90% confidence level.

In addition to addressing the major comments above:

• it is necessary to improve the indication of significance on the map figures in the manuscript. The shading is difficult to see in many of the figures, both on my screen and on a printout.

The indication of significance in all map figures has been modified for improved clarity.

• I would like to see some discussion of how these results compare to previous studies assessing the Australian precipitation response to Asian aerosols. Rotstayn et al. (2007) and Fahrenbach et al. (2023) are cited in the introduction, but are absent from the results and discussion sections.

Thank you for your suggestion. We have now incorporated a discussion comparing our findings with previous studies, particularly Rotstayn et al. (2007) and Fahrenbach et al. (2023).

"Rotstayn et al. (2007) and Fahrenbach et al. (2023) investigated the impact of increasing anthropogenic aerosols over Asia during the late 20th century and early 21st century and indicated that enhanced aerosol emissions strengthened the meridional temperature and pressure gradients, intensifying the monsoonal circulation and leading to increased precipitation over Australia. In contrast, our study focuses on the recent reductions in anthropogenic aerosols in China since 2013, showing that the aerosol reductions reversed circulation changes induced by the previous aerosol increases and contributed to drying rather than increased rainfall over Australia. Our study also explores the subsequent impact of aerosol reductions in China on Australian temperature. We find that reduced moisture availability over Australia led to a shift in surface energy partitioning, with more energy being converted into sensible heat rather than latent heat, resulting in near-surface warming. In summary, while previous studies have demonstrated the role of increasing Asian aerosols in enhancing Australia's precipitation, our work complements these findings by showing that decreasing aerosols, particularly from China, can drive the opposite effect, contributing to both drying and warming conditions in Australia."

The text above has been added to the Conclusion and discussion section.

**References**

Zhai, S., Jacob, D. J., Wang, X., Shen, L., Li, K., Zhang, Y., Gui, K., Zhao, T., and Liao, H.: Fine particulate matter ($PM_{2.5}$) trends in China, 2013–2018: separating contributions from anthropogenic emissions and meteorology, Atmos. Chem. Phys., 19, 11031–11041, https://doi.org/10.5194/acp-19-11031-2019, 2019.

**Responses to Referee #1:**

This research article investigates the impact of anthropogenic aerosol reductions in China on Australia's climate. The study found that the decline in Chine's aerosols since 2013 contributed to drier and warmer conditions in Australia by altering temperature and pressure gradients, which intensified the Southern Trade Winds and caused moisture divergence over Australia. The study also links these climate changes to an increase in wildfire risks in Australia. This research highlights the significant influence of distant aerosols on regional climate and offers insights for drought and wildfire risk mitigation.

The manuscript is interesting, well written and tackles an important topic of research (i.e., impact of Chinese aerosols on Australian climate). However, some technical details between the comparison of modelling results and observations need to be corrected and the selection of figures should be adjusted. I recommend acceptance of the manuscript if the major comment below can be addressed.

We thank the reviewer for the constructive suggestions, which are very helpful for improving the clarity and reliability of the manuscript. Please see our point-by-point responses (in blue) to your comments below.

Major comments

One of my main comments is related to the comparison of observation/reanalysis data and simulated results: There seems to be some inconsistency between the timeperiods used. In the method section it is mentioned that the period 2013-2019 is used for the observation/reanalysis data as well as the simulated data. However, in the captions of the supplementary figures as well as in the description of these figures in the text (e.g. L274, 278) it is mentioned that the observations are for 2010-2019. Please clarify if the same timeperiod is used for observation/reanalysis data and modelling data and if that is not the case, the plots have to be redone for the correct timeperiod to ensure an accurate comparison. Besides, is this warming and drying trend over Australia still continuing or why did the authors look at the time period 2013-2019?

The warming and drying trends in Australia due to the reduction of aerosols in China were simulated based on the two simulations with anthropogenic emissions of aerosols and precursors at years 2013 and 2019. It is because China implemented the "Air Pollution Prevention and Control Plan" in 2013 and established nationwide $PM_{2.5}$ observation sites in 2013. Whether the simulated climate responses can be detected in the real world requires the observational evidence. However, the seven years of period 2013–2019 are too short to fully capture the trends in climate variables, as various natural variabilities could influence temperature and rainfall in Australia. For example, climate variabilities, such as ENSO, can have significant impacts on the Australian monsoon and rainfall patterns and influence the long-term trends of temperature and precipitation in Australia. Therefore, we included more years to calculate the trends of the climate variables in observations. It will not affect the results since that the aerosol reductions in China only account a small amount of the changing climate in Australia. The trends in observations are calculated during 2010–2019 to minimize the internal variability. Regarding whether the trend is continuing, CSIRO and BOM (2022) reported that the warming is still ongoing in Australia and the drying trend is still ongoing in Northern Australia, which could be contributed by the further reductions in Asian aerosols.

While it is great that the author's tried to reduce the figures in the main text to only 4 to explain the whole story, in particular the mechanistic analysis (Section 3.2) is difficult to follow for the reader with the limited number of figures. For instance, a combination of Figure S16 and S17 (i.e. the filled contours showing the SST pattern overlaid by the climatological wind field) would be a relevant figure to show. Additionally, Figure S21 is heavily referenced in the manuscript but the figure is only shown in the supplementary. Besides, maybe a small schematic of the described mechanism similarly as in Fahrenbach et al. 2024 would be helpful to guide the reader through the description.

As requested, we have moved several figures in the main text and added the schematic diagram as shown below.

[Figure]

**Figure 6. Schematic of the response in large-scale 3D circulations in the Asian-Pacific region to aerosol reductions in China.** The top panel shows climatology mean condition, and the bottom panel shows anomalies resulting from aerosol reductions in China.

On the topic of figures, it would be important to show a comparison of the simulated changes with the observed precipitation pattern (Figure S8) as well as the observed wind changes (Figure S18). This is particularly relevant since the authors are trying to do an "attribution" study and it has to be quantified that the observed and modelled changes agree. Additionally, the authors claim that the modelled and observed wind changes are similar (L319-321). While I do acknowledge that 3D wind changes are not the most reliable fields in reanalysis data, this is a bit of an overstatement. Figure S18b and c show very few significant changes making it difficult to understand the simulated flow and Figure S18d shows the largest significant trends in the winds east of Borneo and around stronger southern Trade winds based on the simulated data. Maybe the authors could
think about showing all wind vectors and colouring the significant ones in, so that the reader can
at least see if the observations show the same trend even if they might not be significant based on
this test?

Thank you for your suggestion. We have revised the figure displaying the 3D wind fields, with
significant and insignificant circulations clearly distinguished. Although only a few significant
changes persist in Northern Australia, the large-scale circulations around Australia show noticeable
similarities to the simulated results.

[Figure]

**Figure R1. Linear trends in observed vertical circulations and 850 hPa wind fields in Asia-Pacific regions.** Panel **b** and **c** shows pressure–longitude and latitude cross-section of linear trends in annual mean atmospheric circulations (unit: m s$^{-1}$, vectors) over areas marked with the blue and red box in panel **a** during 2010–2019 from ERA5. Panel **d** shows linear trends of wind fields (unit: m s$^{-1}$, vectors) at 850 hPa in Asia-Pacific regions. Trends of atmospheric circulations and winds which are statistically significant at the 90% confidence level are shown in black, while the insignificant ones are shown in grey.

My last comment regarding the figures is that the figure S15 should also be included in the main
text. It seems biased to try to find a link / attribution but only show the plots for China which the
authors have identified as the relevant one. Maybe a figure showing the annual precipitation trends
for CHN, OTH, NA+EU and then a seasonal plot for the CHN plots would be best?

As requested, we have moved the figure in the main text and showed the CHN, OTH, NA+EU
(Figure 1 below) and then a seasonal plot for the CHN (Figure 2 below).

[Figure]

**Figure 1. Simulated changes in precipitation rate, surface air temperature and relative**
**humidity in Australia due to aerosol changes between 2013 and 2019.** Spatial distributions of
simulated differences in annual mean precipitation rate (Pr, **a**, **d,** and **g**, unit: mm day$^{-1}$), surface
air temperature (TS, **b**, **e**, and **h**, unit: °C) and relative humidity (RH, **c**, **f**, and **i**, unit: %) in
Australia between BASE and CHN (CHN minus BASE, **a**–**c**), between BASE and OTH (OTH
minus BASE, **d**–**f**), and between BASE and NAEU (NAEU minus BASE, **g**–**i**). The shaded areas
indicate results are statistically significant at the 90% confidence level. Regional averages over
Australia are noted at the bottom-left corner of each panel.

[Figure]

**Figure 2. Simulated changes in precipitation rate, surface air temperature and relative humidity in Australia due to aerosol changes in China between 2013 and 2019.** Spatial distributions of simulated differences in DJF (December, January and February, **a**–**c**), MAM (March, April and May, **d**–**f**), JJA (June, July and August, **g**–**i**) and SON (September, October and November, **j**–**l**) mean precipitation rate (Pr, **a**, **d**, **g**, and **j**, unit: mm day$^{-1}$), surface air temperature (TS, **b**, **e**, **h**, and **k**, unit: °C) and relative humidity (RH, **c**, **f**, **i**, and **l**, unit: %) in Australia between BASE and CHN (CHN minus BASE).The shaded areas indicate results are statistically significant at the 90% confidence level. Regional averages over Australia are noted at the bottom-left corner of each panel.

The authors discuss the influence of the (very strong) low bias in PM2.5 in CESM1 compared to the observations in L385-388, which is good and relevant. However, this should also be mentioned throughout the manuscript, for instance when the authors try to estimate very precise values for the influence of the Chinese aerosol reductions on precipitation and temperature (L270-271).

Thank you for your suggestion. We have made some corrections. While there is a low bias in PM$_{2.5}$ in CESM1 compared to observations, this is largely due to the model's inherently low background concentrations. Given that CESM1 has a relatively strong aerosol forcing due to strong aerosol-cloud interactions, the simulated climate effects may not have been significantly underestimated.

Minor comments

L29-31: The times mentioned in this sentence seem confusing since when first reading it seems that a trend from 2013 is caused by something happening around the 2010s. Maybe using
"conditions since the 2010s" would help to settle this confusing sentence.

Revised.

L68-70: Please change "increasing GHGs" to increasing GHG emissions.

Changed.

L72-75: This sentence is very long and confusing, please split it up into two or shorten it

This sentence has been shortened as: "Atmospheric aerosols are the second-largest anthropogenic
climate forcer, exerting an overall cooling effect that partially masks the warming induced by
GHGs."

L73: "Earth's" instead of "earth's"

OK.

L104: "especially in northern Australia/especially in the North of Australia" instead of "especially
the northern Australia"

Revised.

L104: "affected by the Australian monsoon" instead of "affected by Australian monsoon"

Revised.

L153-159: Is there a reason for the choice of the GPM dataset rather than for instance GPCP data?

"GPM provides higher temporal and spatial resolution data compared to GPCP, making it more
suitable for studies focused on short-term precipitation variability and regional climate dynamics."
The statement has been added to the manuscript.

The GPCP data also exhibit a similar decreasing trend in precipitation over Australia, reinforcing
the robustness of the observed trend (Figure S14).

[Figure]

**Figure S14. Linear trends of observed precipitation rate in Australia based on ERA5, GPM,**

**and GPCP.** Spatial distributions of linear trends annual mean precipitation rate (unit: mm day$^{-1}$)
in Australia during 2010–2019 from ERA5 (a), GPM (b), and GPCP (c) datasets. The shaded areas
indicate trends are statistically significant at the 90% confidence level. Regional averages over
Australia are noted at the bottom-left corner of panels.

L224: "Earth's surface" instead of "earth's surface"

Revised.

L244: The setting of DF to 10 according to Sharples et al 2009 needs some more explanation. At
least one sentence why Sharples et al choose this value and why it is also applicable here.

Thank you for your suggestion. Sharples et al. (2009) mentioned that "Such a factor will have no
real bearing on the methods of comparison employed in the later sections of the paper and so for
convenience we assume that DF = 10 in what follows." We acknowledge that this assumption is
idealized. To address this, we have calculated gridded DFs (Figure S7 below) and found that the
DFs in Australia are close to 10, with spatial distributions being nearly homogeneous. Therefore,
setting DF = 10 for Australia is reasonable in this context.

[Figure]

**Figure S9. Spatial distributions of annual mean dry factor (unit: unitless) in Australia during**
**2010–2019.** The data is from fire danger indices historical data from the Copernicus Emergency
Management Service (CEMS, 2019; Vitolo et al., 2020).

Additionally, we compare FFDI with DF = 10 and FFDI with the gridded DF (Figure S8 below).
The patterns and regional averages of both are similar.

[Figure]

**Figure S10. Spatial distribution of simulated changes in FFDI (unit: unitless) during fire**
**seasons in Australia between BASE and CHN (CHN minus BASE).** Shaded areas indicate
results that are statistically significant at the 90% confidence level. Regional averages for Australia are noted at the bottom-left corner of each panel. The left panel shows FFDI (DF = 10), and the
right panel shows FFDI (gridded DF).

L277: Please change "evidence" to "indication".

Changed.

L389: Please use "Earth System Model" or "fully-coupled climate model" instead of "aerosol-
climate model" which would imply to me that this model is not fully coupled (which is the case
according to the method section)

Thanks for your reminder. The term "aerosol-climate model" has been replaced with "fully-
coupled climate model" throughout the manuscript, as suggested.

Figure S3: The colourbar of these two plots should be the same as the reader might be tricked into
thinking that the magnitude changes between the observed and modelled data are similar.

Thank you for your suggestion. If we use the same color scale for both plots, the color range of
the modeled results becomes overly uniform. However, we have now added a note in the figure
caption to remind the reader that the magnitudes of the observed and modeled data are not directly
comparable, and that the color scales represent different ranges.


**Responses to Referee #2:**

The manuscript investigates the role of the recent decline in East Asian anthropogenic aerosols in driving precipitation and temperature changes across Australia. The authors highlight the potential for aerosols to contribute to enhancing the fire risk via widespread drying and warming. The topic is timely and very important, as not many studies have examined the influence of recent changes in aerosols on recent climate variations. However, I find several major weaknesses in the study, including in the simulation design, that prevent the manuscript from being acceptable for ACPD. Unfortunately, I need to recommend a rejection. Happy though to reconsider the manuscript if reviewed accordingly.

Thank you for your feedback. We will carefully consider the points you raised regarding the simulation design and other aspects of the study. We value your insights and will make every effort to improve the quality of our work. We would be grateful for your reconsideration of the manuscript. Please see our point-by-point responses (in blue) to your comments below.

Major comments:

I believe there is a serious overinflation of the recent observed rainfall changes over Australia. Firstly, the authors use observations and plot linear trends from 2010, while model results are shown as 3-year differences from 2013. While the latter accounts for the decline in aerosols, observations show that the years 2010 and 2011 were anomalously wet over Australia, and just a couple of years before rainfall was much less (I have checked by plotting observations since 2000). This can also be inferred by examining Fig 2, where clearly the trend in panel d is affected by the two early years. By eye, using 3-year composites from 2013, the rainfall changes are very modest. Looking at more recent years, it turns out that 2019 was also anomalously dry, and more recent years (excluding those affected by COVID-related reduced emissions) show a recovery. Therefore the results based on observations, which ultimately is the motivation of the study, are strongly affected by the extremely short record and cannot be trusted.

Thank you for pointing out the strong variability in rainfall in Australia. We agree with reviewer that the observed trend of rainfall could be affected by the wet years in 2010/2011 and dry year in 2019, which could be induced by the interval variability (such as ENSO). However, we could not deny the decreasing precipitation during 2010–2019 and rule out the potential influence of changing aerosols on the rainfall variation. Additionally, we are trying to quantify the role of aerosol reduction in China in the Australian climate, rather than fully attribute to changing Australian climate to aerosols in China, considering that aerosol reductions in China are found to contribute to small part of the changing precipitation in Australia. The simulation results are still robust enough to support our conclusions because our model experiments are based on validated climate model, providing strong evidence that aerosol reductions in China have a significant impact on Australia's climate.

Additionally, to account for the potential influence of internal variability, particularly ENSO, which is a major driver of Australia's precipitation variability (Fig. S13), we have also adjusted the precipitation data by removing the influence of the Niño 3.4 index. The adjusted results exhibit a similar trend, confirming the robustness of the observed drying trend since the 2010s (Fig. S12)

[Figure]

**Figure S12. Linear trends of observed precipitation rate during 2001–2019 based on ERA5.** The time series of annual mean precipitation rate (unit: mm day$^{-1}$) over Australia during 2001–2019 from ERA5 reanalysis is shown. The linear trends for the periods 2001–2010 and 2010–2019 are indicated. The trends are also detrended from the Niño 3.4 index.

[Figure]

**Figure S13. Scatter plot showing the correlation between SST anomaly in Niño 3.4 region and precipitation in Australia.** The red line represents the linear regression fit to the data, with the corresponding correlation coefficient (R) and p-value displayed in the figure.

In fact, the emission reduction of aerosols and precursors in China starts in 2010 rather than 2013. We chose the year 2013 in simulation was because China implemented the "Air Pollution Prevention and Control Plan" in 2013 and established nationwide PM$_{2.5}$ observation sites in 2013. As the reviewer mentioned, Australia experienced a transient from dry years before 2010 to wet years in 2010/2011 and the rainfall recovery after 2020. It is also in accordance with the increasing aerosols in China before 2010 and the slowdown in aerosol reduction in recent years. But it still requires quantitative analysis.

Related to the point above, it is not surprising that observed circulation trends (e.g., Fig. S18) are extremely weak. Yet, this is the mechanism driving Australian rainfall changes, thus is central to the proposed aerosol influence.

As mentioned earlier, given natural variability and the multiple factors influencing precipitation in
Australia, it is quite challenging to capture patterns in the observations that resemble the results of
the sensitivity experiment on China's aerosol reductions. We have redrawn the circulation figure
(Figure R1 below), displaying all circulations regardless of whether they pass the significance test.
Notably, we can observe clear similarities with the model results.

[Figure]

**Figure R1. Linear trends in observed vertical circulations and 850 hPa wind fields in Asia-
Pacific regions.** Panel **b** and **c** shows pressure–longitude and latitude cross-section of linear trends
in annual mean atmospheric circulations (unit: m s$^{-1}$, vectors) over areas marked with the blue and
red box in panel **a** during 2010–2019 from ERA5. Panel **d** shows linear trends of wind fields (unit:
m s$^{-1}$, vectors) at 850 hPa in Asia-Pacific regions. Trends of atmospheric circulations and winds
which are statistically significant at the 90% confidence level are shown in black, while the
insignificant ones are shown in grey.

The model analysis is also weak in the sense that 3-year composites are extremely short to identify
forced trends. In addition, observed trends should be first compared to those from a control
simulation and the contribution of aerosols should be related to the model world, not observations.

We believe there may be a misunderstanding regarding the 3-year composites mentioned by the
reviewer. In our study, we did not use 3-year composites for climate variables (only for PM$_{2.5}$ and
AOD in observations since they are more linear than climate variables). Our equilibrium
experiment designs have been described in the 2.2 Model Description and Experimental Design
section.

Regarding the model comparison, we would like to clarify that due to the nature of our equilibrium experiments, direct comparison with baseline control experiments is not feasible. Even in transient simulations, models may not be able to fully capture the short-term and complex observational signal, as observational data are influenced by numerous factors over short time periods. However, models can provide a qualitative evaluation of how specific factors, such as aerosols, might contribute to climate phenomenon. This approach has been commonly used in previous studies (Bollasina et al., 2011; Heede and Fedorov, 2021; Hwang et al., 2024; Vecchi et al., 2006; Wang et al., 2023; Yang et al., 2022) to isolate the impact of individual forcers.

While our simulations provide valuable insights into the physical mechanisms linking aerosol reductions to climate changes, we must admit that they cannot be intended for direct quantitative attribution of observed trends. **To clarify it, we have removed all statements that directly attribute the observed trend to the simulated aerosol response in a quantitative manner and now emphasize that the simulations serve as a tool for understanding the underlying processes rather than quantitatively estimating real-world aerosol-induced changes.** Moving forward, we recognize that transient simulations would be better suited for capturing short-term forced trends, and, we plan to incorporate them in future studies to further investigate the evolution of short-term responses and assess the potential contributions of other external forcings, such as greenhouse gases and internal variability.

Another major weakness is the experimental design. The authors examine long-term changes coming from equilibrium runs, which are far from representing the actual transient response over 10 years or so. This is likely far from being in equilibrium. I would have seen large-ensemble transient simulations with time-varying aerosols (2013-2024 for example) as more appropriate and suitable.

We appreciate the reviewer's concern regarding the use of equilibrium runs and the suggestion to use large-ensemble transient simulations. While transient simulations offer a more dynamic representation of temporal changes, the relatively short period from 2013 to the present may not provide enough time for the climate system to fully respond to aerosol changes. This is particularly true when considering the long timescales required for oceanic heat capacity to equilibrium. Furthermore, equilibrium simulations allow for a clearer isolation of the forced response to aerosol changes, minimizing the impact of short-term climate variability.

We do acknowledge, however, that transient simulations would offer a more accurate depiction of the time-varying effects of aerosol reductions, especially as aerosol reductions in China continue. As the climate system responds more robustly over time, transient simulations will likely become a more appropriate tool. Therefore, while equilibrium simulations are potentially more suitable for the present study, we plan to use transient simulations in future research.

We have also mentioned the limitation of not using transient simulations in the discussion section: "One limitation of this study is the use of equilibrium experiments. Although the equilibrium method is useful for isolating aerosol-induced climate responses, it does not fully capture the transient climatic evolution. Attributing observed climate changes directly to modeled aerosol-induced responses involves inherent uncertainties. Given the uncertainties, here we did not attempt to quantitatively attribute the observed climate change to aerosols in this study, but showed the mechanisms that how aerosol decline influence remote climate. Future work could also consider transient simulations, which can better represent the role of aerosols in the climatic evolution of the real world.".

Trends or changes should always be plotted with the related statistical significance, such as in Figs. 1 and 2.

Thank you for the suggestion. Now the indication of significance in all map figures has been adjusted for improved clarity.

Fig 3: I think it would be more appropriate to display the divergent wind, rather than the total wind.

We chose to display the total wind in Figure 3 to highlight the strengthening of Southern Trade Winds. For the divergent winds, we also displayed moisture divergence (Figure S18), which is more directly related to the changes in precipitation patterns that we are investigating.

**Manuscript with tracked changes:**

[revised manuscript text omitted]

---

## Author Response (AR4)

**Response to Editor:**

There is a nice experiment, and some nice analysis of it in this paper, but there are still major concerns about the framing of the work and the interpretation of the results, and we cannot proceed to publication until these are resolved. It might be that they cannot be resolved. In which case, the authors should motivate their study based on the large emission changes, rather than the observed precipitation, and frame it as a speculative study of the potential responses. However, the issues with interpretation raised by all reviewers must be addressed regardless.

Thank you for your valuable suggestion. We will revise the framing of the manuscript to present it as a speculative study exploring potential climate responses based on the large emission change. We will also ensure that all issues raised by the reviewers are thoroughly addressed. Please see our detailed point-by-point responses (in blue) regarding framing, analysis, and interpretation below.

Framing

The premise of the paper is that Australia has experienced anomalously dry and warm conditions since the 2010. The authors then hypothesise that aerosol reductions in China since 2013 have contributed to this.

We have revised the manuscript to motivate the study based on the large emission changes rather than the observed precipitation, and frame it as a speculative study exploring the potential climate responses throughout the manuscript. Please check the revised paper.

The observed precipitation trend is based on only 10 years of data, and there is still no discussion of how this trend compares to internal variability. The additional Figure S12, provided in response to my comments, raises more concerns here than it solves. The abstract claims that the period since 2010 was warm and dry in Australia, but the mean of the two periods highlighted in Figure S12 are almost identical. There is no evidence in that figure that Australia was anomalously dry since 2010.

We also performed additional analysis examining different time periods, including the 2013–2019 period (Figure S21) and a longer period (2010–2023; Figure S22). Both patterns consistently show a decreasing trend in Australia's precipitation, supporting the robustness of the recent drying trend in Australia, even considering the potential influence of internal variability.

We apologize for the confusion caused by our wording "anomalously dry and warm conditions" in the Abstract. Our intended meaning is that Australia experienced drier and warmer trend since 2010, referring to relative changes within the 2010s, rather than implying that the conditions in 2010s were anomalously dry and warm compared to the long-term historical average. We have revised all similar descriptions.

Figure S14, provided in response to my comments, is reassuring, as it shows that

multiple datasets agree that there has been a drying since 2010. However, I disagree with the authors' conclusion that Figure S12 'clearly shows an increasing precipitation trend during 2001-2010, followed by a decreasing trend during 2010-2019.' What I see here is variability. To be convinced that the trend since 2010 is anomalous, it needs to be compared to all other 10-year trends in ERA5 (using the available data from 1940 to the present to evaluate this). Is the negative trend from 2010-2019 unusually large in the context of 1940-2019?

We agree that variabilities contribute significantly to the trends, and therefore we have removed the previous description stating that Figure S12 "clearly shows an increasing precipitation trend during 2001–2010, followed by a decreasing trend during 2010–2019." As suggested, we have calculated all 10-year precipitation trends in ERA5 from 1940 onwards, as follows (unit: mm day$^{-1}$ year$^{-1}$): +0.016 (1940–1949), –0.012 (1950–1959), +0.001 (1960–1969), +0.031 (1970–1979), –0.002 (1980–1989), +0.030 (1990–1999), –0.032 (2000–2009) and –0.086 (2010–2019). These results indicate that the drying trend during 2010–2019 is unusually large in Australia since 1940 from the ERA5 data.

Figure S12 also highlights the dependence of the 2010-2019 trend on what appears to be anomalously high precipitation in 2010 and 2011. Without these years, the trend would be markedly reduced. This is important, and more so in this case, since the driver that is being explored in the study occurs over 2013-2019. The authors should address the concerns from Reviewer 1 about the sensitivity of the trend to these years, particularly in light of Figure S12.

As per the reviewer's suggestion, we have replotted the spatial distribution of precipitation trends specifically for 2013–2019 (Figure S21). The updated results show an overall decreasing trend across most regions of Australia during this period (except for some localized increases in the northeastern corner). Notably, most regions in northern Australia, which are mainly affected by the trade winds, exhibit even stronger decreases in precipitation during 2013–2019 compared to 2010–2019.

Analysis

The authors perform an equilibrium experiment to examine the Australian response to emission changes in China, North America and Europe, and everywhere except China. This shows that the response to Chinese emission changes between 2013 and 2019 is larger than that to emission changes in other regions. It also shows that the pattern of the precipitation response to Chinese emission changes is different to the response to emission changes elsewhere. The authors also show a strong seasonal component to the response. They also discuss the mechanisms for the response.

Neither the reviewers nor I have substantial comments on the analysis at this stage, as the points raised in review have focused on the framing and interpretation of the study.

The precipitation responses to emission changes in other regions are primarily influenced by emission changes in South Asia and Southeast Asia. The mechanisms of their impacts on Australia's precipitation are generally similar to those of China's

emission reductions. Regarding the strong seasonal component observed in the responses, there are two main contributing factors: (1) the background seasonality of the Southern Hemisphere trade winds, and (2) the seasonality of China's emission changes themselves. However, determining which of these factors dominates requires further detailed quantitative analysis, which we acknowledge as an area for future investigation.

Interpretation

The authors claim that the warming and drying seen in response to Chinese emission reductions in their equilibrium experiments accounts for some of the drying and warming seen in the real world.

Reviewer 1 is concerned about the use of an equilibrium experiment to interpret real-world changes that have happened over a 7-year period. They point to literature that shows that the fast response of Australian precipitation to Asian aerosols can have the opposite sign response to the slow response, and therefore caution that an equilibrium experiment cannot be used to interpret the drivers of a fast change in the real world. It is important that the authors address these concerns.

We have addressed these issues by conducting additional experiments designed specifically to examine the fast climate responses. These new experiments were run for 30 years, with the last 15 years analyzed using the CESM atmospheric component (CAM5) with fixed sea surface temperature. The results show a decreasing pattern in Australian precipitation (Figure S23), which is consistent with the responses obtained from the 150-year equilibrium simulations. This indicates that the precipitation changes in Australia resulting from China's emission reductions are largely contributed by fast responses. We have included these new results and their interpretation in the revised manuscript to address the reviewer's concerns.

There are also some inconsistencies and errors in the response to my comments on the interpretation of the experiment. The authors correctly note that 'the relatively short period from 2013 to the present may not provide enough time for the climate system to fully respond to aerosol changes'. There then needs to be some discussion of the differences between the fast and the slow responses to aerosol changes. As the climate system may not have had time to fully respond to the aerosol changes, how do we expect an equilibrium simulation (where this has happened) to compare to the real world (where it may not have happened)? Can the authors demonstrate that the equilibrium response can be reached within the observational period, so that we can use this experiment to interpret real-world changes? If not, can they comment in detail on where they might expect their experiment to differ from the real-world response? The perturbation is applied as a step change in the equilibrium experiment, so it should be possible to find the time taken for the system to reach equilibrium in this case, and to assess whether the short-term response differs to the equilibrium response. Reviewer 1 also suggests an additional experiment to examine the fast response explicitly. This is a relatively cheap simulation compared to the equilibrium runs, and would only need to be performed for the China experiment.

We have added detailed discussions in the revised manuscript to clarify the differences between fast and slow responses and the consistency between our fast response and long-term equilibrium experiments. We also acknowledge that while equilibrium simulations provide insights into the long-term full response, fast response experiments are more appropriate for attribution of short-term observed trends, and we have reframed our study accordingly to explore potential climate responses based on the large emission changes rather than direct attribution.

"Both fast and slow climate responses contribute to the simulated precipitation changes in Australia resulting from China's emission reductions. Fast responses are primarily driven by rapid atmospheric adjustments to aerosol-induced radiative forcing without requiring full ocean adjustment, while slow responses involve gradual changes mediated by ocean circulation and sea surface temperature adjustments over longer timescales.

In our additional fast response experiments using the CAM5 atmospheric component, we found a decreasing pattern in Australia's precipitation (Figure S23), which is consistent with the long-term equilibrium simulation results. Although the equilibrium simulations represent the fully adjusted climate system response and are not appropriate to be directly compared to short-term observational changes, the fast response experiments suggest that fast response accounts 86% of the fully climate system response in the changing precipitation in Australia. It demonstrates that the immediate atmospheric response to Chinese aerosol reductions alone can produce a drying effect similar to that found in long-period equilibrium experiments, providing confidence in the robustness of the results."

The authors have removed most direct attribution statements from the text, but Reviewer 2 highlights one that remains, and should also be removed.

We have carefully reviewed the manuscript again and removed the remaining attribution statement highlighted by Reviewer 2.

**Response to Reviewer#1:**

The manuscript now includes more relevant in the main and supplementary material. However, the manuscript still has some major caveats related to the experimental setup and observational comparison after the comments from the reviewers and editor. I recommend another round of major revision based on the comments below. If the authors would not be able to give a scientific reasoning why a comparison of the equilibrium simulations with the transient (fast) observed precipitation response over Australia (see major comment 1) is valid, I would recommend that the article is rejected but could be considered for resubmission if the author's performed simulations to assess the fast climate response and find that these simulations justify their hypothesis.

Thank you very much for your overall evaluation and comments. We have provided detailed point-by-point responses (in blue) to each of your comments below and revised the manuscript accordingly to address these remaining concerns.

Major comments

1. My main concern is still regarding the experimental setup since the changes over Australia in an equilibrium climate simulation (around 100 years) are compared to the fast climate response in observations (less than 10 years). A paper by Liu et al 2018 examined the fast and slow precipitation response over Australia shows that while the fast response shows a drying to Asian sulfate aerosols, the slow response shows a wettening (see Figure 1). Additionally, a recent paper by Hwang et al 2024 shows very different east-west Pacific and Indian Ocean SST patterns which lead to differences in the flow towards Australia (see Figure 3). Based on this previous literature, it does not seem justified to attribute the recent short-term drying (fast response) in Australia using equilibrium simulations which only show the slow responses.

In our new experiments designed to additionally examine the fast climate responses to aerosol reductions in China, which were run for 30 years with the last 15 years analyzed using the CESM atmospheric component (CAM5), we also found a decreasing pattern in Australian precipitation (Figure S23). This is consistent with the total long-term equilibrium responses, suggesting that the precipitation changes in Australia resulting from China's emission reductions are largely contributed by fast responses. Moreover, in some regions where the changes pass significance tests, such as northern and eastern Australia, the precipitation decrease in the fast response experiments is even larger than that in the fully-coupled simulations. It demonstrates that the immediate atmospheric response to Chinese aerosol reductions alone can produce a drying effect similar to that found in long-period equilibrium experiments,

providing confidence in the robustness of the results.

We note that Figure 1 of Liu et al. (2018) shows statistically insignificant precipitation changes over Australia in both the fast and slow responses to Asian sulfate aerosols. In addition to the drying induced by Chinese emission reduction, we also evaluated the influence of emission increases in South Asia and Southeast Asia and found aerosol increases in these regions could reduce precipitation over Australia (Figure 1j). Liu et al. (2018) examined emissions from the entire Asian region, and therefore, the precipitation signals over Australia were not significant. Hwang et al. (2024) focused on aerosol changes at the global scale rather than on emission changes in a specific region. Both their study did not specifically assess how the fast and slow climate responses would manifest in response to emission decreases from China alone.

It should also be noted that we did not attempt to directly explain the recent short-term drying in Australia using the more than 100-year equilibrium model runs. The equilibrium experiments were designed to explore the potential climate effects of China's aerosol emission reductions on Australia and to investigate the associated mechanisms by which these emission reductions could influence Australian precipitation. While the fast responses provide more relevant insights for explaining short-term changes, the long-term equilibrium experiments remain valuable for understanding the potential long-term impacts, especially considering that emission reductions are expected to continue in the future.

In response to the reviewer and editor the authors write the following:

- o "While transient simulations offer a more dynamic representation of temporal changes, the relatively short period from 2013 to the present may not provide enough time for the climate system to fully respond to aerosol changes."

- o "As the climate system responds more robustly over time, transient simulations will likely become a more appropriate tool."

I would argue that both these statements are not correct: In the former, if the authors argue that the "real world climate" did not have enough time to fully respond to the aerosol changes then the same reasoning should be applied to the modelled climate. Thus, it seems incorrect to compare equilibrium climate simulations where the climate system had a lot of time to fully respond to the aerosol changes with the transient "real world climate". Similarly, the second comment is incorrect as over time the equilibrium simulations will become more appropriate (e.g. if it is a long-term Australian drying trend that should be attributed) while the short-term trend that the authors try to assess would be more accurately captured by a fast response.

Thanks for pointing out these improper expressions. As noted in our previous response, we did not attempt to directly explain the recent short-term drying in Australia using equilibrium simulations with more than 100 years. The equilibrium experiments in this study were designed to explore the potential influence of China's emission reductions on Australia's precipitation and to investigate the underlying physical mechanisms, rather than to attribute the recent short-term observed changes. This study is not intended as an attribution analysis and we have now reframed the manuscript to avoid the misleading. Another purpose of using long-term equilibrium simulations is to assess whether the precipitation changes induced by China's emission reductions would persist under continued emission reduction scenarios in the future. Moreover, we have also added atmosphere-only simulations to quantify the fast response in precipitation to the aerosol reductions in China. For the second point, we decide to withdraw our previous response.

The authors now added a short paragraph in the discussion (L432-439). However, the authors will have to add a detailed discussion of their results (and choice to use an equilibrium simulations) in the light of the papers by Hwang et al 2024 and Liu et al 2018 as well as any other relevant papers. How can this attribution of the fast Australian drying based on equilibrium simulations be trusted if previous literature shows large differences in the precipitation patterns over Australia and SST around Australia (and related mechanisms) in the fast and slow response? If the authors would be unable to give a scientific reasoning, I would recommend that the article is rejected but could be considered for resubmission if the author's performed simulations to assess the fast climate response and find that these simulations justify their hypothesis.

Thank you for your comment. As noted earlier, this study is not intended as an attribution analysis. Regarding the studies by Hwang et al. (2024) and Liu et al. (2018), we would like to clarify their findings and relevance to our work.

"Liu et al. (2018) used multi-model simulations to investigate the fast and slow precipitation responses to regional aerosol forcing from Asia and Europe under the PDRMIP framework. They showed that Asian sulfate aerosols were stronger drivers of global temperature and precipitation changes compared to European aerosols. However, their analysis focused on aerosol forcing from the entire Asian region, rather than emission reductions from China alone, and the precipitation responses over Australia were not significant in their results. Hwang et al. (2024) explored the combined fast and slow climate responses to aerosol emission changes at the global scale. They reported that an increase in aerosol emissions, followed by a decrease, can sustain La Niña-like patterns in the equatorial Pacific for decades. However, neither study specifically examined the climate impacts of China's recent emission reductions. We acknowledge that fast and slow climate responses can have different

mechanisms and regional impacts, and there can be differences between the slow and fast responses. Our additional experiments with atmosphere-only indicate that the fast response dominated the drying pattern over Australia (Figure S23), account for 86% of the fully-coupled response, suggesting that the precipitation decrease is mainly driven by the fast response."

We have added a detailed discussion in the revised manuscript to highlight these findings and clarify the differences in experimental focus and implications.

2. Thanks for providing the additional Figures S12 and S14 which help to examine the effect of different datasets and time periods. However, the authors still focus on the observational 2010-2019 and do not address how the anomalously wet year in 2010/11 might bias their assessment. This is particularly relevant since the authors theoretically want to compare the influence of Chinese aerosols on precipitation trends over Australia from 2013-2019.

In order to assess the impact of including these three additional years, I recommend to create a spatial precipitation trend figure based on observational data showing the 2013-2019 trend in comparison to the 2010-2019 trend that the authors already show. If the 2013-2019 trend plot shows similar changes as the 2010-2019 plot, then this could help to make their statement of including the additional 3 years to reduce the influence of internal variability more robust.

Additionally, the large impact of internal variability in the observational data should be discussed in the discussion further.

We have drawn the spatial distributions of precipitation trends based on observational data for 2013–2019 (Figure S21). The new plot shows a broadly similar drying pattern to the 2010–2019 trend, with widespread decreases in precipitation across northern Australia (except for slight increases in the northeast).

"It should be noted that the short-term precipitation trends examined in this study may be influenced by internal climate variability, especially given the relatively short period since 2013. Therefore, we extended the analysis back to around 2010, when emission reductions in China had already begun. We also performed additional analysis examining different time periods, including the 2013–2019 period (Figure S21) and a longer period (2010–2023; Figure S22). Both patterns consistently show a decreasing trend in Australia's precipitation, supporting the robustness of the recent drying trend in Australia. However, we acknowledge that these observed short-term trends could still be affected by internal climate variability, and the potential contribution from internal variability should not be ignored."

We agree that internal variability can have a large influence on short-term trends. Therefore, we have added the discussion above to the revised manuscript.

Liu, L., et al. (2018): "A PDRMIP multimodel study on the impacts of regional aerosol forcings on global and regional precipitation." *Journal of climate* 31.11, 4429-4447.

Hwang, Yen-Ting, et al. (2024): "Contribution of anthropogenic aerosols to persistent La Niña-like conditions in the early 21st century." *Proceedings of the National Academy of Sciences* 121.5

**Response to Reviewer#2:**

The authors have improved the manuscript in light of the reviewers' comments. However, I am still not fully convinced by the analysis performed and the use of the specific experimental design. While the revised manuscript, as suggested, more cautiously uses the results of the CESM simulations to attribute the recent observed trends, it still relies on the analysis of the long-term century-long output, which is not what happens in the real world (a ~10-year trend). Do the simulated circulation patterns resemble observations? The analysis of the observational record should also extend to more recent years, leading to 15-year time series. Finally, some paragraphs still suggest an attribution-like approach ("In contrast, our study focuses on the recent reductions in anthropogenic aerosols in China since 2013, showing that the aerosol reductions reversed circulation changes induced by the previous aerosol increases and contributed to drying rather than increased rainfall over Australia.").

Thank you for your comments. In our new experiments designed to examine the fast climate responses, we also find a decreasing pattern in Australian precipitation (Figure S23). This is consistent with the total 150-year equilibrium responses, suggesting that the precipitation changes in Australia resulting from China's emission reductions are largely contributed by fast responses. It demonstrates that the immediate atmospheric response to Chinese aerosol reductions alone can produce a drying effect similar to that found in long-period equilibrium experiments, providing confidence in the robustness of the results.

Following the editor's suggestion, we have revised the framing of the manuscript to motivate our study based on the large emission changes in China rather than the observed precipitation trends, presenting it as a speculative study exploring potential climate responses. With this reframing, the purpose of the study is no longer to attribute the observed drying trend in Australia to China's emission reductions but to examine the possible climate impacts of these emission changes on Australia. We believe this revision addresses the core concerns raised regarding attribution.

Regarding the simulated circulation patterns, they show a reasonable resemblance to the observed circulation anomalies. However, under the revised framing, even if the circulation patterns in the model do not exactly match the observations, it does not undermine the scientific value of analyzing the model results, as many factors influence circulation and precipitation variability around Australia, and an exact match with observations is not necessarily expected.

We have also extended the observational precipitation time series to include more recent years as suggested (Figure S22), providing a longer context for the analysis.

Finally, we have revised or removed all paragraphs suggesting an attribution-like interpretation to ensure consistency with the new framing of the manuscript.

[Figure]

**Figure 1. Simulated changes in precipitation rate, surface air temperature and relative humidity in Australia due to aerosol changes between 2013 and 2019.** Spatial distributions of simulated differences in annual mean precipitation rate (Pr, **a**, **d**, and **g**, unit: mm day$^{-1}$), surface air temperature (TS, **b**, **e**, and **h**, unit: °C) and relative humidity (RH, **c**, **f**, and **i**, unit: %) in Australia between BASE and CHN (CHN minus BASE, **a–c**), between BASE and OTH (OTH minus BASE, **d–f**), between BASE and NAEU (NAEU minus BASE, **g–i**), and between BASE and SASEA (SASEA minus BASE, **j–l**). The shaded areas indicate results are statistically significant at the 90% confidence level. Regional averages over Australia are noted at the bottom-left corner of each panel.

[Figure]

**Figure S21. Linear trends of observed precipitation rate in Australia during 2013–2019 based on ERA5.** Spatial distributions of linear trends annual mean precipitation rate (unit: mm day$^{-1}$) in Australia during 2013–2019 from ERA5. The shaded areas indicate trends are statistically significant at the 90% confidence level.

[Figure]

**Figure S22. Linear trends of observed precipitation rate in Australia during 2010–2023 based on ERA5.** Spatial distributions of linear trends annual mean precipitation rate (unit: mm day$^{-1}$) in Australia during 2010–2023 from ERA5. The shaded areas indicate trends are statistically significant at the 90% confidence level.

[Figure]

**Figure S23. Simulated changes in precipitation rate in Australia due to aerosol changes in China between 2013 and 2019.** Spatial distributions of simulated differences in annual mean precipitation rate (unit: mm day$^{-1}$) in Australia between BASE_FAST and CHN_FAST (CHN_FAST minus BASE_FAST). The shaded areas indicate results are statistically significant at the 90% confidence level. These experiments were run for 30 years, with the last 15 years analyzed using the CESM atmospheric component (CAM5) with fixed sea surface temperature.